# Genotype-stratified treatment for monogenic insulin resistance: a systematic review

Robert K. Semple[1,2], Kashyap A. Patel [3,4], Sungyoung Auh[5], ADA/EASD PMDI* & Rebecca J. Brown [5✉]

### Abstract

**Background** Monogenic insulin resistance (IR) includes lipodystrophy and disorders of insulin signalling. We sought to assess the effects of interventions in monogenic IR, stratified by genetic aetiology.

**Methods** Systematic review using PubMed, MEDLINE and Embase (1 January 1987 to 23 June 2021). Studies reporting individual-level effects of pharmacologic and/or surgical interventions in monogenic IR were eligible. Individual data were extracted and duplicates were removed. Outcomes were analysed for each gene and intervention, and in aggregate for partial, generalised and all lipodystrophy.

**Results** 10 non-randomised experimental studies, 8 case series, and 23 case reports meet inclusion criteria, all rated as having moderate or serious risk of bias. Metreleptin use is associated with the lowering of triglycerides and haemoglobin A1c (HbA1c) in all lipodystrophy ($n = 111$), partial ($n = 71$) and generalised lipodystrophy ($n = 41$), and in *LMNA*, *PPARG*, *AGPAT2* or *BSCL2* subgroups ($n = 72, 13, 21$ and $21$ respectively). Body Mass Index (BMI) is lowered in partial and generalised lipodystrophy, and in *LMNA or BSCL2*, but not *PPARG* or *AGPAT2* subgroups. Thiazolidinediones are associated with improved HbA1c and triglycerides in all lipodystrophy ($n = 13$), improved HbA1c in *PPARG* ($n = 5$), and improved triglycerides in *LMNA* ($n = 7$). In *INSR*-related IR, rhIGF-1, alone or with IGFBP3, is associated with improved HbA1c ($n = 17$). The small size or absence of other genotype-treatment combinations preclude firm conclusions.

**Conclusions** The evidence guiding genotype-specific treatment of monogenic IR is of low to very low quality. Metreleptin and Thiazolidinediones appear to improve metabolic markers in lipodystrophy, and rhIGF-1 appears to lower HbA1c in INSR-related IR. For other interventions, there is insufficient evidence to assess efficacy and risks in aggregated lipodystrophy or genetic subgroups.

### Plain language summary

The hormone insulin stimulates nutrient uptake from the bloodstream into tissues. In insulin resistance (IR), this action is blunted. Some rare gene alterations cause severe IR, diabetes that is difficult to control, and early complications. Many treatments have been suggested, but reliable evidence of their risks and benefits is sparse. We analysed all available reports describing treatment outcomes in severe IR. We found that the evidence is of low to very low quality overall. Injections of leptin, a hormone from fat tissue, or thiazolidinedione tablets that increase fat tissue both appear to improve diabetes control in people with reduced ability to make fat tissue. Injections of another treatment, insulin-like growth factor, appear to improve diabetes control in people with direct blockage of insulin action. There is a pressing need to improve evidence for treatment in these rare and severe conditions.

[1] Centre for Cardiovascular Science, Queen's Medical Research Institute, University of Edinburgh, Edinburgh, UK. [2] MRC Human Genetics Unit, Institute of Genetics and Cancer, University of Edinburgh, Edinburgh, UK. [3] Department of Clinical and Biomedical Sciences, University of Exeter Medical School, Exeter, UK. [4] Department of Diabetes and Endocrinology, Royal Devon and Exeter NHS Foundation Trust, Exeter, UK. [5] National Institute of Diabetes and Digestive and Kidney Diseases, National Institutes of Health, Bethesda, MD, USA. *A list of authors and their affiliations appears at the end of the paper.
✉email: brownrebecca@niddk.nih.gov

Diabetes caused by single gene changes is highly hetero-geneous in molecular aetiopathogenesis. It may be grouped into disorders featuring primary failure of insulin secretion, and disorders in which insulin resistance (IR), often severe, predates secondary failure of insulin secretion and diabetes. Monogenic IR is itself heterogeneous, encompassing primary lipodystrophy syndromes, primary disorders of insulin signalling, and a group of conditions in which severe IR is part of a more complex developmental syndrome[1].

Monogenic IR is rare but underdiagnosed. The commonest subgroup is formed by genetic lipodystrophy syndromes[2,3]. A recent analysis of a large clinical care cohort unselected for metabolic disease suggested a clinical prevalence of lipodystrophy of around 1 in 20,000, with a prevalence of plausible lipodystrophy-causing genetic variants of around 1 in 7000[4]. Monogenic IR is important to recognise, because affected patients are at risk not only of micro- and macrovascular complications of diabetes, but also of complications such as dyslipidemia, pancreatitis, and steatohepatitis, especially in lipodystrophy syndromes[5]. Non-metabolic complications specific to individual gene defects may also occur, including hypertrophic cardiomyopathy and other manifestations of soft tissue overgrowth[3]. Diabetes is also commonly the sentinel presentation of a multisystem disorder, and recognition of complex syndromes in a diabetes clinic may trigger definitive diagnostic testing.

The only therapy licensed specifically for monogenic IR is recombinant human methionyl leptin (metreleptin), with licensed indications encompassing a subset of patients with lipodystrophy and inadequate metabolic control. The current license in the USA is restricted to generalised lipodystrophy, but in Europe, it extends to some patients with partial lipodystrophy. A substantial proportion of the body of evidence considered in licensing addressed patients ascertained by the presence of clinical lipodystrophy, and the role of genetic stratification in the precision treatment of lipodystrophy has not been systematically addressed. Many other medications and other treatment options are also widely used in monogenic IR, although not licensed for that specific subgroup. Such use draws on the evidence base and treatment algorithms developed for type 2 diabetes. Several forms of monogenic IR have molecular and/or clinical attributes that suggest potential precision approaches to treatment.

We sought now to undertake a systematic review of the current evidence guiding the treatment of monogenic IR stratified by genetic aetiology, to assess evidence for differential responses to currently used therapies, to establish gaps in evidence, and to inform future studies. This systematic review is written on behalf of the American Diabetes Association (ADA)/European Association for the Study of Diabetes (EASD) *Precision Medicine in Diabetes Initiative* (PMDI) as part of a comprehensive evidence evaluation in support of the 2nd International Consensus Report on Precision Diabetes Medicine[6]. The PMDI was established in 2018 by the ADA in partnership with the EASD to address the burgeoning need for better diabetes prevention and care through precision medicine[7].

Our analyses show that metreleptin and thiazolidinediones appear to lower HbA1c, triglycerides, and body weight in patients with lipodystrophy of all genotypes, and rhIGF-1 appears to lower HbA1c in patients with *INSR*-related IR. For other interventions, there is insufficient evidence to assess efficacy and risks.

## Methods
**Inclusion criteria and search methodology**. To assess the treatment of severe IR of known monogenic aetiology, with or without diabetes mellitus, including generalised and partial lipodystrophy and genetic disorders of the insulin receptor, we

developed, registered and followed a protocol for a systematic review (PROSPERO ID CRD42021265365; registered July 21, 2021)[8]. The study was reported in accordance with Preferred Reporting Items for Systematic Reviews and Meta-Analysis (PRISMA) guidelines. Filtering and selection of studies for data extraction were recorded using the Covidence platform (https://www.covidence.org, Melbourne, Australia).

We searched PubMed, MEDLINE and Embase from 1987 (the year before the identification of the first monogenic aetiology of IR) to June 23, 2021 for potentially relevant human studies in English. We used broad search terms designed to capture the heterogeneity of monogenic IR and its treatments. We searched for studies addressing 1. Severe IR due to variant(s) in a single gene OR 2. Congenital generalised or familial partial lipodystrophy due to variant(s) in a single gene. We selected only studies that reported a treatment term, including but not limited to the mention of 1. Thiazolidinediones (TZD), 2. Metreleptin, 3. SGLT2 inhibitors, 4. GLP-1 analogues, 5. Bariatric surgery (all types), 6. Recombinant human IGF-1 or IGF-1/IGFBP3 composite, 7. U-500 insulin. No interventions were excluded in the primary search. In addition to the automated search, we hand searched reference lists of relevant review articles. Given the rarity of monogenic IR, no study types were excluded in the initial search. We ultimately considered experimental studies, case reports, and case series. The full search strategy is described in Supplementary Table 1.

Study selection for data extraction was performed in two phases, namely primary screening of title and abstract, then full text review of potentially eligible articles. Two authors independently evaluated eligibility, with discrepancies resolved by a third investigator. We excluded publications without original data, such as reviews, editorials, and comments, and those solely addressing severe IR or lipodystrophy of unknown or known non-monogenic aetiology, including HIV-related or other acquired lipodystrophies, or autoimmune insulin receptoropathy (Type B insulin resistance). Studies in which no clear categorical or numerical outcome of an intervention was reported, or in which interventions were administered for less than 28 days were also excluded.

**Data extraction and outcome assessments**. One author extracted data from each eligible study using data extraction sheets. Data from each study was verified by 3 authors to reach a consensus. Data were extracted from text, tables, or figures. Study investigators were contacted for pertinent unreported data or additional details where possible, most commonly genetic aetiology of insulin resistance in reported patients, and outcome data.

Data extracted for each study included first author, publication year, country, details of intervention, duration of follow-up, study design, and number of participants. Subject-level data were extracted for outcomes of interest, including sex, genetic cause of severe insulin resistance (gene name, mono- vs biallelic *INSR* pathogenic variant), phenotypic details of severe IR/lipodystrophic subtype (generalised vs partial lipodystrophy; associated syndromic features). Subject level outcome data were extracted prior to and after the longest-reported exposure to the intervention of interest for haemoglobin A1c (A1c), body mass index, serum triglyceride, ALT, or AST concentration, any index of liver size or lipid content, and total daily insulin dose. Potential adverse effects of interventions were recorded, including urinary tract infection, genital candidiasis, hypoglycemia, excessive weight loss, pancreatitis, soft tissue overgrowth, and tumour formation.

**Risk of bias and certainty of evidence assessment**. The quality of extracted case reports and case series was assessed using NIH

Study Quality Assessment Tools[9] by a single reviewer and verified by 2 additional reviewers. Grading of the overall evidence for specific research questions was undertaken as detailed in[10].

**Statistics and reproducibility**. Extracted data were managed using Covidence and analysed with SAS version 9.4. Pooled analysis was undertaken for all combinations of genotype and intervention for which sufficient numbers were reported, as well as for aggregated lipodystrophies, and generalized and partial subgroups of lipodystrophy. Generalized Estimating Equation models were used with time as a fixed factor and study as a random factor to examine treatment effects. Serum triglyceride concentrations were analyzed with and without log transformation. Data were summarized using estimated least-squared means with corresponding 95% confidence intervals.

**Reporting summary**. Further information on research design is available in the Nature Portfolio Reporting Summary linked to this article.

## Results

**Identification of eligible studies**. Initial searching identified 2933 studies, to which 117 were added from the bibliography reviews. 256 articles remained after the screening of titles and abstracts, and 44 after full text screening (Fig. 1).

**Included studies addressed limited interventions and most had a high risk of bias**. The 44 studies analysed and the assessment of their quality are summarised in Table 1 and detailed in Supplementary Data 1. Study quality was assessed as being fair in 15 cases and poor in 29 cases, including all case reports. This was primarily due to the high risk of bias, particularly related to the lack of a control group for all studies. Three of the 44 studies included in further analysis included only individuals already described in other reports and were discarded, leaving 41 studies for final analysis. These comprised 10 non-controlled experimental studies, 8 case series and 23 individual case reports (Table 1). No controlled trials were found. Individuals reported in the studies included 90 with partial lipodystrophy (72 due to *LMNA* mutation and 15 due to *PPARG* mutation), 42 with generalized lipodystrophy (21 *AGPAT2*, 21 *BSCL2*, 2 *LMNA*), and 19 with IR due to *INSR* mutation(s). Among the interventions described, only the responses to metreleptin (111 recipients), thiazolidinediones (13 recipients) and rhIGF-1 (alone or as a composite with IGFBP3) (17 recipients) were described in more than 5 cases (Table 1). This meant that for the large preponderance of possible genotype-treatment combinations no specific data were recovered (Supplementary Table 2). Full outcome data extracted are summarised in Supplementary Data 2, and subject-level data are shown in Supplementary Figures 1 through 8 with raw data provided in Supplementary Data 2.

**Metreleptin treatment was associated with improved metabolic control in lipodystrophy**. In our registered systematic review plan we posed several subquestions about treatment of monogenic IR subtypes that we felt were tractable. The first related to the risks and benefits (assessed by side effects, A1c, serum triglyceride concentration, body mass index (BMI), and indices of fatty liver) of metreleptin in patients with different monogenic subtypes of lipodystrophy. The response to metreleptin was

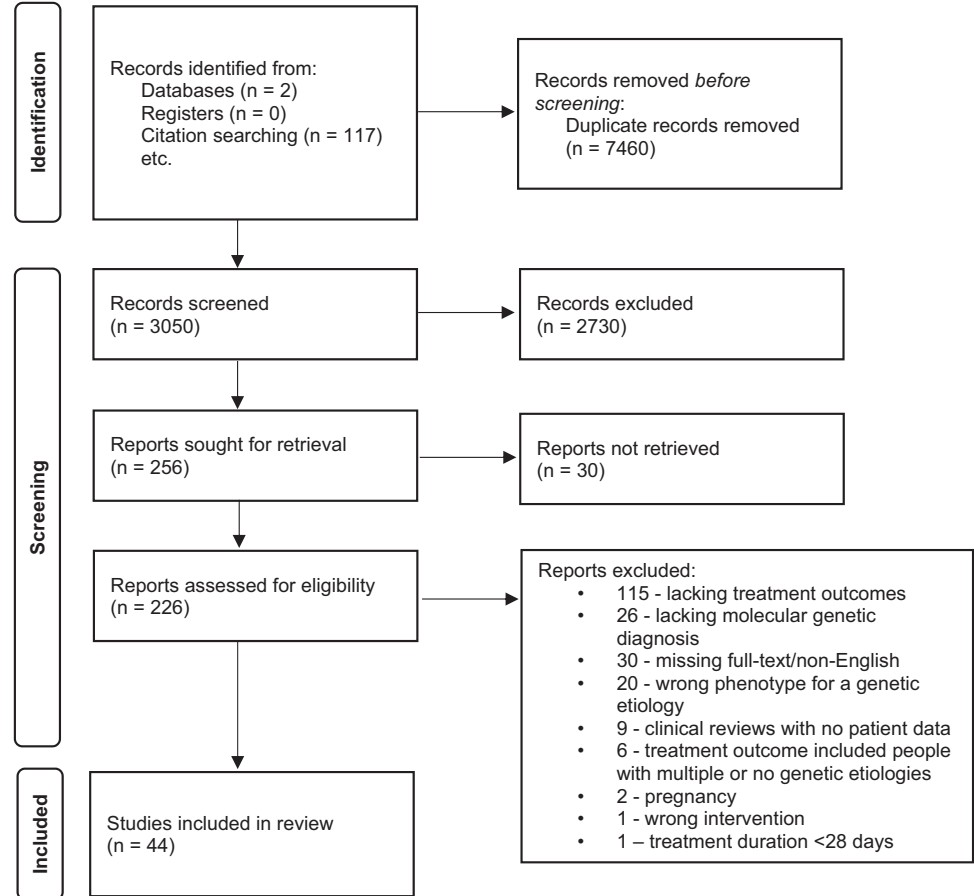

**Fig. 1 PRISMA diagram.** PRISMA flow diagram of publications evaluated based on the search strategy.

**Table 1 Summary characteristics of included studies.**

| Study types | Number of studies |
| --- | --- |
| Case reports | 23 |
| Non-randomised experimental study | 10 |
| Case series | 8 |
| **Study Quality\*** | **Number of studies** |
| Good | 0 |
| Fair | 15 |
| Poor | 30 |
| **Phenotypes** | **Number of participants** |
| Partial lipodystrophy | 90 |
| | (72 *LMNA*, 15 *PPARG*, 2 *PLIN1*, 1 *PIK3R1*) |
| Generalised lipodystrophy | 56 |
| | (21 *AGPAT2*, 21 *BSCL2*, 1 *PTRF*, 2 *LMNA*) |
| Insulin receptor | 19 (7 Monoallelic, 12 Biallelic) |
| **Intervention** | **Number of participants#** |
| Metreleptin | 111 (71/40/0) |
| rhIGF-1 or rhIGF-1/IGFBP3 composite | 17 (0/0/17) |
| Thiazolidinedione | 13 (12/1/0) |
| Metformin | 5 (2/1/2) |
| Bariatric surgery | 4 (4/0/0) |
| SGLT2i | 2 (1/1/0) |

\*Based on NHLBI quality assessment tool; #Numbers in brackets are for partial lipodystrophy/ generalised lipodystrophy/ insulin receptor individuals respectively. *rhIGF-1* recombinant human insulin-like growth factor 1, *IGFBP3* insulin-like growth factor binding protein 3, *SGLT2i* sodium-glucose co-transporter-2 inhibitor

described in 111 people (71 with partial lipodystrophy, 40 with generalized lipodystrophy)[11–23]. Metreleptin was administered for 19 ± 20 months (median 12, range 1–108) and was associated with lowering of A1c in aggregated lipodystrophy, in generalized and partial subgroups, and in all genetic subgroups for whom sufficient patients were reported, namely those with *LMNA*, *PPARG*, *AGPAT2* and *BSCL2* mutations (0.5 to 1.5% least square mean reduction) (Level 3 evidence, Supplementary Data 3, Fig. 2). Metreleptin treatment was also associated with lowering of serum triglyceride concentration in aggregated lipodystrophy, in generalized and partial subgroups, and in those with *LMNA*, *PPARG*, *AGPAT2* and *BSCL2* mutations (92 to 1760 mg/dL least square mean reduction for analyses of untransformed data) (Level 3 evidence, Supplementary Data 3, Fig. 2). BMI was lower after treatment in aggregated lipodystrophy, in generalized and partial subgroups, and in those with *LMNA* or *BSCL2* mutations, but not *PPARG* or *AGPAT2* mutations (Level 3 evidence, Supplementary Data 3, Fig. 2). Liver outcomes reported were too heterogeneous to analyse in aggregate. Only a single adverse event, namely hypoglycemia, was reported.

**Thiazolidinedione treatment showed variable efficacy in limited studies.** We next addressed the evidence of the risks and benefits of thiazolidinediones (TZDs) in patients with lipodystrophy. We were specifically interested in any evidence of a greater or lesser response in partial lipodystrophy caused by *PPARG* variants than in other lipodystrophy subtypes, as TZDs are potent ligands for the product of the *PPARG* gene, the master regulator of adipocyte differentiation. The response to TZDs was described in only 13 people, however (12 FPLD, 1 CGL)[24–34]. TZDs were administered for 29 ± 28 months (median 24, range 2–96). TZD use was associated with improved A1c in aggregated lipodystrophy (least square mean reduction 2.2%) and in *PPARG*-related but not *LMNA*-related partial lipodystrophy (Level 4 evidence,

Supplementary Data 3, Fig. 3). Serum triglyceride concentration decreased in aggregated lipodystrophy and in those with *LMNA*-related but not *PPARG*-related partial lipodystrophy (Level 4 evidence, Supplementary Data 3, Fig. 3). No adverse events were reported.

**rhIGF-1 treatment in INSR-related IR was associated with improvement in A1c.** Our last specific question related to the risks (e.g. tumours, hypoglycemia, cardiac hypertrophy, other soft tissue overgrowth) and benefits (assessed by A1c) of recombinant human IGF-1 (rhIGF-1) or IGF-1/IGFBP3 composite in patients with pathogenic *INSR* variants. The response to rhIGF-1 was described in 17 people with pathogenic *INSR* variants for a mean of 45 ± 81 months (median 9, range 1–288)[35–46]. In *INSR*-related IR, we found that use of rhIGF-1, alone or as a composite with IGFBP3, was associated with improvement in A1c, and this was true also in subgroups with monoallelic and biallelic variants (1.5 to 2% least square mean reduction, Level 4 evidence, Supplementary Data 3, Fig. 4). One instance of increased soft tissue overgrowth and two episodes of hypoglycemia was reported.

**Many questions about genotype-stratified treatment were not addressed.** While many other interesting and clinically relevant questions arise about other potential genotype-specific responses to therapy in monogenic IR, the small size or absence of other genotype by treatment groups precluded the drawing of conclusions about risks and benefits, including for very widely used medications such as metformin[26,47–49], newer agents commonly used in type 2 diabetes including SGLT2 inhibitors[50,51] and GLP-1 agonists, and non pharmacologic interventions such as bariatric surgery[52–54].

## Discussion

Thirty-five years since *INSR* mutations were identified in extreme IR[55,56], and 23 years since the first monogenic cause of lipodystrophy was reported[57], many different forms of monogenic IR are known[1–3,58]. These are associated with substantial early morbidity and mortality, ranging from death in infancy to accelerated complications of diabetes and fatty liver disease in adulthood, depending on the genetic subtype. Several opportunities for genotype-guided, targeted treatment are suggested by the causal genes, and so we set out to review the current evidence guiding the treatment of monogenic IR stratified by genetic aetiology. We found a paucity of high-quality evidence (all levels 3 to 4). No controlled trials of any intervention were identified, and there was substantial heterogeneity of study populations and intervention regimens, even for the same interventional agent.

The evidence which we did find, from a small number of uncontrolled experimental studies, augmented by case series and numerous case reports, suggests that metreleptin offers metabolic benefits across different lipodystrophy subtypes, in keeping with its licensing for use in some patients with lipodystrophy in both Europe and the USA. Notably, the evidence base considered by licensing authorities was larger than the one we present, including many studies of phenotypically ascertained lipodystrophy that included acquired or idiopathic disease. In contrast, we have addressed solely individuals with lipodystrophy caused by variation in a single gene. The limited data we identified do not clearly support differential effects among different monogenic lipodystrophy subgroups, but for many subtypes numbers reported are very small. Moreover, although responses appear comparable for partial and generalised lipodystrophy, this is highly likely to reflect selection bias in studies of partial lipodystrophy towards those with more severe metabolic complications and lower baseline serum leptin concentrations.

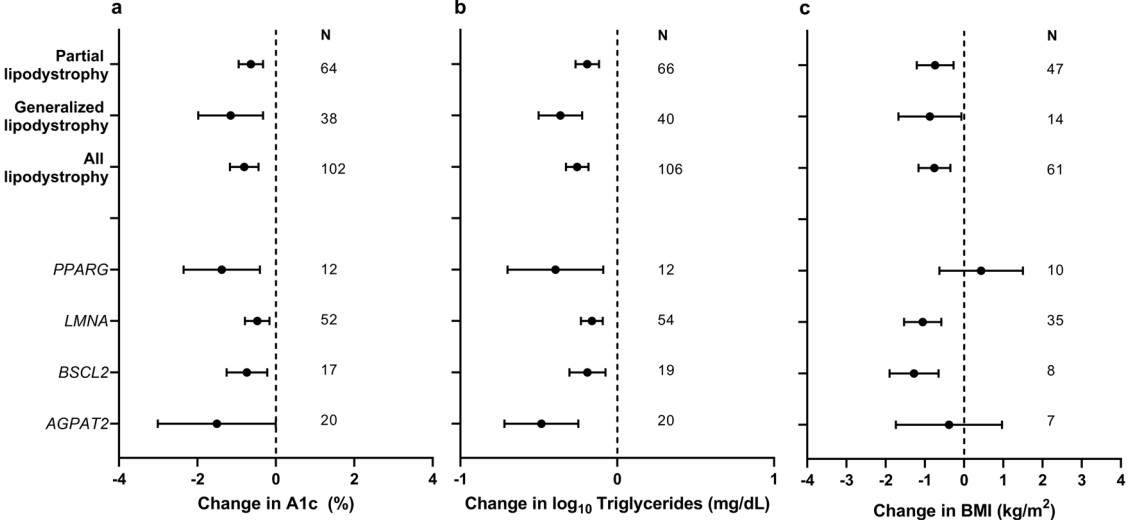

**Fig. 2 Effects of metreleptin in monogenic forms of lipodystrophy.** Least square mean change in (**a**) Hemoglobin A1c (A1c), (**b**) Log$_{10}$ serum triglyceride concentration and (**c**) Body Mass Index (BMI) in patients with partial lipodystrophy, generalized lipodystrophy, all forms of lipodystrophy, and subgroups with *PPARG*, *LMNA*, *BSCL2*, and *AGPAT2* mutations. Error bars represent 95% confidence intervals. N = 64, 38, 102, 12, 52, 17, and 20 for change in A1c in partial lipodystrophy, generalized lipodystrophy, all lipodystrophy, *PPARG*, *LMNA*, *BSCL2*, and *AGPAT2*-associated lipodystrophy, respectively. N = 66, 40, 106, 12, 54, 19, and 20 for change in log$_{10}$ triglycerides in partial lipodystrophy, generalized lipodystrophy, all lipodystrophy, *PPARG*, *LMNA*, *BSCL2*, and *AGPAT2*-associated lipodystrophy, respectively. N = 47, 14, 61, 10, 35, 8, and 7 for change in BMI in partial lipodystrophy, generalized lipodystrophy, all lipodystrophy, *PPARG*, *LMNA*, *BSCL2*, and *AGPAT2*-associated lipodystrophy, respectively.

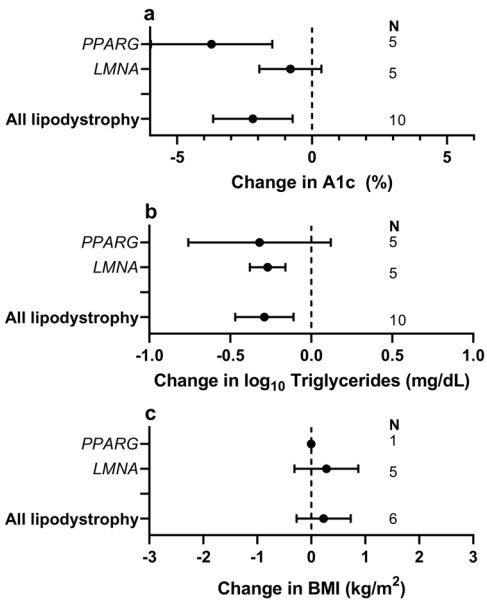

**Fig. 3 Effects of thiazolidinediones in monogenic forms of lipodystrophy.** Least square mean change in (**a**) Hemoglobin A1c (A1c), (**b**) Log$_{10}$ serum triglyceride concentration and (**c**) Body Mass Index (BMI) in patients with partial lipodystrophy, generalized lipodystrophy, all forms of lipodystrophy, and subgroups with *PPARG*, and *LMNA* mutations. Error bars represent 95% confidence intervals. N = 5, 5, and 10 for change in A1c and change in log$_{10}$ triglycerides in *PPARG*, *LMNA*, and all lipodystrophy, respectively. N = 1, 5, and 6 for change in BMI in *PPARG*, *LMNA*, and all lipodystrophy, respectively.

A clear opportunity for precision diabetes therapy in monogenic IR is offered by the IR and lipodystrophy caused by mutations in *PPARG*, which encodes the target for thiazolidinediones (TZDs) such as pioglitazone[59,60]. PPARG is a nuclear receptor that serves as the master transcriptional driver of adipocyte differentiation, and so as soon as *PPARG* mutations were

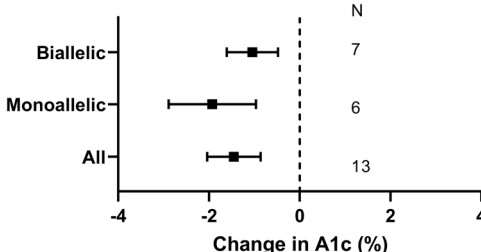

**Fig. 4 Effects of recombinant human Insulin-like Growth Factor-1 (rhIGF) alone or in combination with Insulin-like Growth Factor Binding Protein-3 (IGFBP3) in patients with *INSR* mutations.** Least square mean change in hemoglobin A1c (A1c), in all patients with *INSR* mutations, and in subgroups with biallelic and monoallelic mutations. Error bars represent 95% confidence intervals. N = 7, 6, and 13 for biallelic, monoallelic, and all *INSR* mutations.

identified to cause severe IR, there was interest in the potential of TZDs as specific treatments. Although we found small scale evidence supporting greater A1c reduction with TZDs in *PPARG* vs. *LMNA*-related lipodystrophy, only 5 patients with *PPARG*-related lipodystrophy in whom TZD effects were clearly described were reported, and responses were inconsistent. Thus, it remains unclear whether people with IR due to *PPARG* variants are more or indeed less sensitive to TZDs than people with other forms of lipodystrophy. Loss-of-function *PPARG* mutations are the second commonest cause of familial partial lipodystrophy[2], and the function of coding missense variants in *PPARG* has been assayed systematically to accelerate genetic diagnosis[61], so the opportunity to test genotype-related therapy in *PPARG*-related IR seems particularly tractable in future.

Other obvious questions about the targeted treatment of monogenic, lipodystrophic IR are not addressed by current evidence. Important examples relate to the risks and benefits of treatments used in type 2 diabetes such as GLP-1 agonists and SGLT2 inhibitors. It is rational to suppose that these medications, which decrease weight as well as improve glycaemia in those with

raised BMI and diabetes, may also be efficacious in lipodystrophy even where BMI is normal or only slightly raised. This is because in both situations adipose storage capacity is exceeded, leading to fat failure. It is the offloading of overloaded adipose tissue, rather than the baseline BMI/adipose mass, which underlies the efficacy of therapy. However, GLP-1 agonists are contraindicated in those with prior pancreatitis, while SGLT2 inhibitor use can be complicated by diabetic ketoacidosis. In untreated lipodystrophy pancreatitis is common, yet this is due to hypertriglyceridaemia, which is likely to be improved by GLP-1 agonist use, while excessive supply of free fatty acids to the liver may promote ketogenesis. Thus, assessment of both classes of drugs in lipodystrophy and its genetic subgroups will be important to quantify risks and benefits, which may be distinct from those in obesity-related diabetes.

A further question we prespecified related to the use of rhIGF1 in people with severe IR due to *INSR* mutations. This use of rhIGF-1 was first described in recessive *INSR* defects in the early 1990s[44], and several studies of rhIGF-1 therapy of duration less than 28 days in people with *INSR* mutations have provided proof of concept for acute metabolic benefits (summarized in[38]). This use of rhIGF-1 is based on the rationale that IGF-1 activates a receptor and signalling pathway very closely similar to those activated by insulin. Based on case reports, case series and narrative reviews, rhIGF-1 is now commonly used in neonates with extreme IR due to biallelic *INSR* mutations, although, unlike metreleptin in lipodystrophy, this use is still unlicensed. Our review of published data, which was limited to durations of intervention greater than 28 days, is consistent with glycaemic benefits of rhIGF-1, alone or in composite form with its binding protein IGFBP3, in people with *INSR* mutations. Nevertheless, such studies are challenging to interpret and are potentially fraught with bias of different types, particularly publication bias favouring positive outcomes. Responses to rhIGF1 are also challenging to determine in uncontrolled studies as small differences in the residual function of mutated receptors can have substantial effects on the severity and natural history of the resulting IR, yet relatively few *INSR* mutations have been studied functionally. This underlines the narrow nature of, and substantial residual uncertainty in, the evidence base for the use of rhIGF-1 in monogenic IR.

There are several reasons why important questions about the precision treatment of monogenic IR have not been settled. Although severe autosomal recessive IR is usually detected in infancy, commoner dominant forms of monogenic IR are often diagnosed relatively late, often only after years of management based on presumptive diagnoses of type 2 or sometimes type 1 diabetes. Initial management as type 2 diabetes means that by the time a clinical and then genetic diagnosis is made, most patents have been treated with agents such as metformin, and increasingly SGLT2 inhibitors or GLP-1 agonists, outside trial settings. It is not clear that harm is caused by such use of drugs with well-established safety profiles and efficacy in type 2 diabetes, but the lack of systematic data gathering precludes the identification of specific drug-genotype interactions. Moreover, because attempts to gather evidence for monogenic IR treatment have tended to focus on high-cost adjunctive therapies such as metreleptin, the evidence base for their use is better developed, although controlled trials are lacking. Licensing of high-cost treatments such as metreleptin in lipodystrophy, while effects of many more commonly used, cheaper drugs with well-established safety profiles lack formal testing in monogenic IR is potentially problematic, skewing incentives and guidelines towards expensive therapy before optimal treatment algorithms have been established.

Other challenges in conducting trials in monogenic IR arise from the exquisite sensitivity of IR to exacerbating factors such as puberty, diet, and energy balance. This creates a signal to noise problem particularly problematic in uncontrolled studies, in which non-pharmacological components of interventions such as increased support for behavioural change may confound attribution of beneficial outcomes to pharmacological agents tested.

The key question now is how the evidence base for managing monogenic severe IR can be improved in the face of constraints in studying rare, clinically heterogeneous, and geographically dispersed patients who are often diagnosed late with a condition that is exquisitely environmentally sensitive. Growing interest in and development of methodologies for clinical trials in rare disease[62], including Bayesian methodologies[63,64], and hybrid single- and multi-site designs[65] offer hope for future filling of evidence gaps. One important and pragmatic opportunity arises from the development of large regional, national and international networks and registries for lipodystrophy (e.g. the Europe-based ECLip registry[66]), allied to emergence of randomised registry-based trial (RRT) methodology[67,68]. RRTs have attracted increasing interest in several disease areas and are particularly suitable for evaluation of agents with well-established safety profiles. When a simple randomisation tool is deployed in the context of a registry, RRTs can offer rapid, cost-effective recruitment and high external validity (i.e. relevance to real world practice). In monogenic IR this would permit questions to be addressed about optimal usage of different common medications in different genetic subgroups, including the order of introduction of therapies, and their optimal combinations. The quality of such studies will critically rely on good registry design and quality and completeness of data capture[67,68].

In summary, severe monogenic IR syndromes are clinically and genetically heterogeneous, with high early morbidity and mortality. However, despite opportunities for targeted therapy of some monogenic subgroups based on the nature of the causal gene alteration, the evidence for genotype-stratified therapy is weak. This is in part because of the rarity and frequent late diagnosis of monogenic IR, but also because therapeutic research to date has focused largely on phenotypically ascertained cross cutting diagnoses such as lipodystrophy. We suggest that approaches such as RRTs hold the best hope to answer some of the persisting major questions about precision treatment in monogenic IR.

## Data availability
All data used in this review is available from publicly available and herein referenced sources. A list of included studies is provided in Supplementary Data 1. All data generated or analyzed during this study are included in this published article and its supplementary information files. Source data for the figures are available as Supplementary Data 2.

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

## Acknowledgements
This research was funded in part, by the Wellcome Trust [Grant WT 210752 to RKS and WT 219606 to KAP]. For the purpose of open access, the author has applied a CC0 Public Domain Dedication to any Author Accepted Manuscript version arising from this submission. RJB and SA are supported by the intramural research program of the National Institute of Diabetes and Digestive and Kidney Diseases. The ADA/EASD Precision Diabetes Medicine Initiative, within which this work was conducted, has received the following support: The Covidence license was funded by Lund University (Sweden) for which technical support was provided by Maria Björklund and Krister Aronsson (Faculty of Medicine Library, Lund University, Sweden). Administrative support was provided by Lund University (Malmö, Sweden), University of Chicago (IL, USA), and the American Diabetes Association (Washington D.C., USA). The Novo Nordisk Foundation (Hellerup, Denmark) provided grant support for in-person writing group meetings (PI: L Phillipson, University of Chicago, IL).

## Author contributions
R.K.S., R.J.B., and K.A.P. researched data, wrote the manuscript, and reviewed and approved the final manuscript. S.A. conducted statistical analyses and reviewed and approved the final manuscript. Members of the ADA/EASD PMDI Consortium provided feedback on methodology and reporting guidelines.

## Competing interests
The authors declare the following competing interests: R.K.S. has received speaker fees from Eli Lilly, Novo Nordisk, and Amryt. R. J. B. has received research support from Amryt, Third Rock Ventures, Ionis, and Regeneron. K.A.P. and S.A. report no conflicts of interest.

## Additional information

## ADA/EASD PMDI

Deirdre K. Tobias[6,7], Jordi Merino[8,9,10], Abrar Ahmad[11], Catherine Aiken[12,13], Jamie L. Benham[14], Dhanasekaran Bodhini[15], Amy L. Clark[16], Kevin Colclough[3], Rosa Corcoy[17,18,19], Sara J. Cromer[9,20,21], Daisy Duan[22], Jamie L. Felton[23,24,25], Ellen C. Francis[26], Pieter Gillard[27], Véronique Gingras[28,29], Romy Gaillard[30], Eram Haider[31], Alice Hughes[3], Jennifer M. Ikle[32,33], Laura M. Jacobsen[34], Anna R. Kahkoska[35], Jarno L. T. Kettunen[36,37,38], Raymond J. Kreienkamp[9,10,20,39], Lee-Ling Lim[40,41,42], Jonna M. E. Männistö[43,44], Robert Massey[31], Niamh-Maire Mclennan[1], Rachel G. Miller[45], Mario Luca Morieri[46,47], Jasper Most[48], Rochelle N. Naylor[49], Bige Ozkan[50,51], Kashyap Amratlal Patel[3], Scott J. Pilla[52,53], Katsiaryna Prystupa[54,55], Sridaran Raghaven[56,57], Mary R. Rooney[50,58], Martin Schön[54,55,59], Zhila Semnani-Azad[7], Magdalena Sevilla-Gonzalez[20,21,60], Pernille Svalastoga[61,62], Wubet Worku Takele[63], Claudia Ha-ting Tam[42,64,65], Anne Cathrine B. Thuesen[8], Mustafa Tosur[66,67,68], Amelia S. Wallace[50,58], Caroline C. Wang[58], Jessie J. Wong[69], Jennifer M. Yamamoto[70], Katherine Young[3], Chloé Amouyal[71,72], Mette K. Andersen[8], Maxine P. Bonham[73], Mingling Chen[74], Feifei Cheng[75], Tinashe Chikowore[21,76,77,78], Sian C. Chivers[79], Christoffer Clemmensen[8], Dana Dabelea[80], Adem Y. Dawed[31], Aaron J. Deutsch[10,20,21], Laura T. Dickens[81], Linda A. DiMeglio[23,24,25,82], Monika Dudenhöffer-Pfeifer[11], Carmella Evans-Molina[23,24,25,83],

María Mercè Fernández-Balsells[84,85], Hugo Fitipaldi[11], Stephanie L. Fitzpatrick[86], Stephen E. Gitelman[87], Mark O. Goodarzi[88,89], Jessica A. Grieger[90,91], Marta Guasch-Ferré[7,92], Nahal Habibi[90,91], Torben Hansen[8], Chuiguo Huang[42,64], Arianna Harris-Kawano[23,24,25], Heba M. Ismail[23,24,25], Benjamin Hoag[93,94], Randi K. Johnson[95,96], Angus G. Jones[3,97], Robert W. Koivula[98], Aaron Leong[9,21,99], Gloria K. W. Leung[73], Ingrid M. Libman[100], Kai Liu[90], S. Alice Long[101], William L. Lowe Jr.[102], Robert W. Morton[103,104,105], Ayesha A. Motala[106], Suna Onengut-Gumuscu[107], James S. Pankow[108], Maleesa Pathirana[90,91], Sofia Pazmino[109], Dianna Perez[23,24,25], John R. Petrie[110], Camille E. Powe[9,20,21,111], Alejandra Quinteros[90], Rashmi Jain[112,113], Debashree Ray[58,114], Mathias Ried-Larsen[115,116], Zeb Saeed[117], Vanessa Santhakumar[6], Sarah Kanbour[52,118], Sudipa Sarkar[52], Gabriela S. F. Monaco[23,24,25], Denise M. Scholtens[119], Elizabeth Selvin[50,58], Wayne Huey-Herng Sheu[120,121,122], Cate Speake[123], Maggie A. Stanislawski[95], Nele Steenackers[109], Andrea K. Steck[124], Norbert Stefan[55,125,126], Julie Støy[127], Rachael Taylor[128], Sok Cin Tye[129,130], Gebresilasea Gendisha Ukke[63], Marzhan Urazbayeva[67,131], Bart Van der Schueren[109,132], Camille Vatier[133,134], John M. Wentworth[135,136,137], Wesley Hannah[138,139], Sara L. White[79,140], Gechang Yu[42,64], Yingchai Zhang[42,64], Shao J. Zhou[91,141], Jacques Beltrand[142,143], Michel Polak[142,143], Ingvild Aukrust[61,144], Elisa de Franco[3], Sarah E. Flanagan[3], Kristin A. Maloney[145], Andrew McGovern[3], Janne Molnes[61,144], Mariam Nakabuye[8], Pål Rasmus Njølstad[61,62], Hugo Pomares-Millan[11,146], Michele Provenzano[147], Cécile Saint-Martin[148], Cuilin Zhang[149,150], Yeyi Zhu[151,152], Sungyoung Auh[153], Russell de Souza[104,154], Andrea J. Fawcett[155,156], Chandra Gruber[157], Eskedar Getie Mekonnen[158,159], Emily Mixter[160], Diana Sherifali[104,161], Robert H. Eckel[162], John J. Nolan[163,164], Louis H. Philipson[160], Rebecca J. Brown[153], Liana K. Billings[165,166], Kristen Boyle[80], Tina Costacou[45], John M. Dennis[3], Jose C. Florez[9,10,20,21], Anna L. Gloyn[32,33,167], Maria F. Gomez[11,168], Peter A. Gottlieb[124], Siri Atma W. Greeley[169], Kurt Griffin[113,170], Andrew T. Hattersley[3,97], Irl B. Hirsch[171], Marie-France Hivert[9,172,173], Korey K. Hood[69], Jami L. Josefson[155], Soo Heon Kwak[174], Lori M. Laffel[175], Siew S. Lim[63], Ruth J. F. Loos[8,176], Ronald C. W. Ma[42,64,65], Chantal Mathieu[27], Nestoras Mathioudakis[52], James B. Meigs[21,99,177], Shivani Misra[178,179], Viswanathan Mohan[180], Rinki Murphy[181,182,183], Richard Oram[3,97], Katharine R. Owen[98,184], Susan E. Ozanne[185], Ewan R. Pearson[31], Wei Perng[80], Toni I. Pollin[145,186], Rodica Pop-Busui[187], Richard E. Pratley[188], Leanne M. Redman[189], Maria J. Redondo[66,67], Rebecca M. Reynolds[1], Robert K. Semple[1,2], Jennifer L. Sherr[190], Emily K. Sims[23,24,25], Arianne Sweeting[191,192], Tiinamaija Tuomi[36,38,136], Miriam S. Udler[9,10,20,21], Kimberly K. Vesco[193], Tina Vilsbøll[194,195], Robert Wagner[54,55,196], Stephen S. Rich[107] & Paul W. Franks[7,11,98,105]

[6]Division of Preventative Medicine, Department of Medicine, Brigham and Women's Hospital and Harvard Medical School, Boston, MA, USA. [7]Department of Nutrition, Harvard T.H. Chan School of Public Health, Boston, MA, USA. [8]Novo Nordisk Foundation Center for Basic Metabolic Research, Faculty of Health and Medical Sciences, University of Copenhagen, Copenhagen, Denmark. [9]Diabetes Unit, Endocrine Division, Massachusetts General Hospital, Boston, MA, USA. [10]Center for Genomic Medicine, Massachusetts General Hospital, Boston, MA, USA. [11]Department of Clinical Sciences, Lund University Diabetes Centre, Lund University Malmö, Sweden. [12]Department of Obstetrics and Gynaecology, the Rosie Hospital, Cambridge, UK. [13]NIHR Cambridge Biomedical Research Centre, University of Cambridge, Cambridge, UK. [14]Departments of Medicine and Community Health Sciences, Cumming School of Medicine, University of Calgary, Calgary, AB, Canada. [15]Department of Molecular Genetics, Madras Diabetes Research Foundation, Chennai, India. [16]Division of Pediatric Endocrinology, Department of Pediatrics, Saint Louis University School of Medicine, SSM Health Cardinal Glennon Children's Hospital, St. Louis, MO, USA. [17]CIBER-BBN, ISCIII, Madrid, Spain. [18]Institut d'Investigació Biomèdica Sant Pau (IIB SANT PAU), Barcelona, Spain. [19]Departament de Medicina, Universitat Autònoma de Barcelona, Bellaterra, Spain. [20]Programs in Metabolism and Medical & Population Genetics, Broad Institute, Cambridge, MA, USA. [21]Department of Medicine, Harvard Medical School, Boston, MA, USA. [22]Division of Endocrinology, Diabetes and Metabolism, Johns Hopkins University School of Medicine, Baltimore, MD, USA. [23]Department of Pediatrics, Indiana University School of Medicine, Indianapolis, IN, USA. [24]Herman B Wells Center for Pediatric Research, Indiana University School of Medicine, Indianapolis, IN, USA. [25]Center for Diabetes and Metabolic Diseases, Indiana University School of Medicine, Indianapolis, IN, USA. [26]Department of Biostatistics and Epidemiology, Rutgers School of Public Health, Piscataway, NJ, USA. [27]University Hospital Leuven, Leuven, Belgium. [28]Department of Nutrition, Université de Montréal, Montreal, QC, Canada. [29]Research Center, Sainte-Justine University Hospital Center, Montreal, QC, Canada. [30]Department of Pediatrics, Erasmus Medical Center, Rotterdam, The Netherlands. [31]Division of Population Health & Genomics, School of Medicine, University of Dundee, Dundee, UK. [32]Department of Pediatrics, Stanford School of Medicine, Stanford University, Stanford, CA, USA. [33]Stanford Diabetes Research Center, Stanford School of Medicine, Stanford University, Stanford, CA, USA. [34]University of Florida, Gainesville, FL, USA. [35]Department of Nutrition, University of North Carolina at Chapel Hill, Chapel Hill, NC, USA. [36]Helsinki University Hospital, Abdominal Centre/Endocrinology, Helsinki, Finland. [37]Folkhalsan Research Center,

Helsinki, Finland. [38]Institute for Molecular Medicine Finland FIMM, University of Helsinki, Helsinki, Finland. [39]Department of Pediatrics, Division of Endocrinology, Boston Children's Hospital, Boston, MA, USA. [40]Department of Medicine, Faculty of Medicine, University of Malaya, Kuala Lumpur, Malaysia. [41]Asia Diabetes Foundation, Hong Kong SAR, China. [42]Department of Medicine & Therapeutics, Chinese University of Hong Kong, Hong Kong SAR, China. [43]Departments of Pediatrics and Clinical Genetics, Kuopio University Hospital, Kuopio, Finland. [44]Department of Medicine, University of Eastern Finland, Kuopio, Finland. [45]Department of Epidemiology, University of Pittsburgh, Pittsburgh, PA, USA. [46]Metabolic Disease Unit, University Hospital of Padova, Padova, Italy. [47]Department of Medicine, University of Padova, Padova, Italy. [48]Department of Orthopedics, Zuyderland Medical Center, Sittard-Geleen, The Netherlands. [49]Departments of Pediatrics and Medicine, University of Chicago, Chicago, IL, USA. [50]Welch Center for Prevention, Epidemiology, and Clinical Research, Johns Hopkins Bloomberg School of Public Health, Baltimore, MD, USA. [51]Ciccarone Center for the Prevention of Cardiovascular Disease, Johns Hopkins School of Medicine, Baltimore, MD, USA. [52]Department of Medicine, Johns Hopkins University, Baltimore, MD, USA. [53]Department of Health Policy and Management, Johns Hopkins University Bloomberg School of Public Health, Baltimore, MD, USA. [54]Institute for Clinical Diabetology, German Diabetes Center, Leibniz Center for Diabetes Research at Heinrich Heine University Düsseldorf, Auf'm Hennekamp 65, 40225 Düsseldorf, Germany. [55]German Center for Diabetes Research (DZD), Ingolstädter Landstraße 1, 85764 Neuherberg, Germany. [56]Section of Academic Primary Care, US Department of Veterans Affairs Eastern Colorado Health Care System, Aurora, CO, USA. [57]Department of Medicine, University of Colorado School of Medicine, Aurora, CO, USA. [58]Department of Epidemiology, Johns Hopkins Bloomberg School of Public Health, Baltimore, MD, USA. [59]Institute of Experimental Endocrinology, Biomedical Research Center, Slovak Academy of Sciences, Bratislava, Slovakia. [60]Clinical and Translational Epidemiology Unit, Massachusetts General Hospital, Boston, MA, USA. [61]Mohn Center for Diabetes Precision Medicine, Department of Clinical Science, University of Bergen, Bergen, Norway. [62]Children and Youth Clinic, Haukeland University Hospital, Bergen, Norway. [63]Eastern Health Clinical School, Monash University, Melbourne, VIC, Australia. [64]Laboratory for Molecular Epidemiology in Diabetes, Li Ka Shing Institute of Health Sciences, The Chinese University of Hong Kong, Hong Kong, China. [65]Hong Kong Institute of Diabetes and Obesity, The Chinese University of Hong Kong, Hong Kong, China. [66]Department of Pediatrics, Baylor College of Medicine, Houston, TX, USA. [67]Division of Pediatric Diabetes and Endocrinology, Texas Children's Hospital, Houston, TX, USA. [68]Children's Nutrition Research Center, USDA/ARS, Houston, TX, USA. [69]Stanford University School of Medicine, Stanford, CA, USA. [70]Internal Medicine, University of Manitoba, Winnipeg, MB, Canada. [71]Department of Diabetology, APHP, Paris, France. [72]Sorbonne Université, INSERM, NutriOmic team, Paris, France. [73]Department of Nutrition, Dietetics and Food, Monash University, Melbourne, VIC, Australia. [74]Monash Centre for Health Research and Implementation, Monash University, Clayton, VIC, Australia. [75]Health Management Center, The Second Affiliated Hospital of Chongqing Medical University, Chongqing Medical University, Chongqing, China. [76]MRC/Wits Developmental Pathways for Health Research Unit, Department of Paediatrics, Faculty of Health Sciences, University of the Witwatersrand, Johannesburg, South Africa. [77]Channing Division of Network Medicine, Brigham and Women's Hospital, Boston, MA, USA. [78]Sydney Brenner Institute for Molecular Bioscience, Faculty of Health Sciences, University of the Witwatersrand, Johannesburg, South Africa. [79]Department of Women and Children's health, King's College London, London, UK. [80]Lifecourse Epidemiology of Adiposity and Diabetes (LEAD) Center, University of Colorado Anschutz Medical Campus, Aurora, CO, USA. [81]Section of Adult and Pediatric Endocrinology, Diabetes and Metabolism, Kovler Diabetes Center, University of Chicago, Chicago, USA. [82]Department of Pediatrics, Riley Hospital for Children, Indiana University School of Medicine, Indianapolis, IN, USA. [83]Richard L. Roudebush VAMC, Indianapolis, IN, USA. [84]Biomedical Research Institute Girona, IdIBGi, Girona, Spain. [85]Diabetes, Endocrinology and Nutrition Unit Girona, University Hospital Dr Josep Trueta, Girona, Spain. [86]Institute of Health System Science, Feinstein Institutes for Medical Research, Northwell Health, Manhasset, NY, USA. [87]University of California at San Francisco, Department of Pediatrics, Diabetes Center, San Francisco, CA, USA. [88]Division of Endocrinology, Diabetes and Metabolism, Cedars-Sinai Medical Center, Los Angeles, CA, USA. [89]Department of Medicine, Cedars-Sinai Medical Center, Los Angeles, CA, USA. [90]Adelaide Medical School, Faculty of Health and Medical Sciences, The University of Adelaide, Adelaide, Australia. [91]Robinson Research Institute, The University of Adelaide, Adelaide, Australia. [92]Department of Public Health and Novo Nordisk Foundation Center for Basic Metabolic Research, Faculty of Health and Medical Sciences, University of Copenhagen, 1014 Copenhagen, Denmark. [93]Division of Endocrinology and Diabetes, Department of Pediatrics, Sanford Children's Hospital, Sioux Falls, SD, USA. [94]University of South Dakota School of Medicine, E Clark St, Vermillion, SD, USA. [95]Department of Biomedical Informatics, University of Colorado Anschutz Medical Campus, Aurora, CO, USA. [96]Department of Epidemiology, Colorado School of Public Health, Aurora, CO, USA. [97]Royal Devon University Healthcare NHS Foundation Trust, Exeter, UK. [98]Oxford Centre for Diabetes, Endocrinology and Metabolism, University of Oxford, Oxford, UK. [99]Division of General Internal Medicine, Massachusetts General Hospital, Boston, MA, USA. [100]UPMC Children's Hospital of Pittsburgh, Pittsburgh, PA, USA. [101]Center for Translational Immunology, Benaroya Research Institute, Seattle, WA, USA. [102]Department of Medicine, Northwestern University Feinberg School of Medicine, Chicago, IL, USA. [103]Department of Pathology & Molecular Medicine, McMaster University, Hamilton, Canada. [104]Population Health Research Institute, Hamilton, Canada. [105]Department of Translational Medicine, Medical Science, Novo Nordisk Foundation, Tuborg Havnevej 19, 2900 Hellerup, Denmark. [106]Department of Diabetes and Endocrinology, Nelson R Mandela School of Medicine, University of KwaZulu-Natal, Durban, South Africa. [107]Center for Public Health Genomics, Department of Public Health Sciences, University of Virginia, Charlottesville, VA, USA. [108]Division of Epidemiology and Community Health, School of Public Health, University of Minnesota, Minneapolis, MN, USA. [109]Department of Chronic Diseases and Metabolism, Clinical and Experimental Endocrinology, KU Leuven, Leuven, Belgium. [110]School of Health and Wellbeing, College of Medical, Veterinary and Life Sciences, University of Glasgow, Glasgow, UK. [111]Department of Obstetrics, Gynecology, and Reproductive Biology, Massachusetts General Hospital and Harvard Medical School, Boston, MA, USA. [112]Sanford Children's Specialty Clinic, Sioux Falls, SD, USA. [113]Department of Pediatrics, Sanford School of Medicine, University of South Dakota, Sioux Falls, SD, USA. [114]Department of Biostatistics, Johns Hopkins Bloomberg School of Public Health, Baltimore, Maryland, USA. [115]Centre for Physical Activity Research, Rigshospitalet, Copenhagen, Denmark. [116]Institute for Sports and Clinical Biomechanics, University of Southern Denmark, Odense, Denmark. [117]Department of Medicine, Division of Endocrinology, Diabetes and Metabolism, Indiana University School of Medicine, Indianapolis, IN, USA. [118]AMAN Hospital, Doha, Qatar. [119]Department of Preventive Medicine, Division of Biostatistics, Northwestern University Feinberg School of Medicine, Chicago, IL, USA. [120]Institute of Molecular and Genomic Medicine, National Health Research Institutes, Taipei City, Taiwan. [121]Divsion of Endocrinology and Metabolism, Taichung Veterans General Hospital, Taichung, Taiwan. [122]Division of Endocrinology and Metabolism, Taipei Veterans General Hospital, Taipei, Taiwan. [123]Center for Interventional Immunology, Benaroya Research Institute, Seattle, WA, USA. [124]Barbara Davis Center for Diabetes, University of Colorado Anschutz Medical Campus, Aurora, CO, USA. [125]University Hospital of Tübingen, Tübingen, Germany. [126]Institute of Diabetes Research and Metabolic Diseases (IDM), Helmholtz Center Munich, Neuherberg, Germany. [127]Steno Diabetes Center Aarhus, Aarhus University Hospital, Aarhus, Denmark. [128]University of Newcastle, Newcastle upon Tyne, UK. [129]Sections on Genetics and Epidemiology, Joslin Diabetes Center, Harvard Medical School, Boston, MA, USA. [130]Department of Clinical Pharmacy and Pharmacology, University Medical Center Groningen, Groningen, The Netherlands. [131]Gastroenterology, Baylor College of Medicine, Houston, TX, USA. [132]Department of Endocrinology, University Hospitals Leuven, Leuven, Belgium. [133]Sorbonne University, Inserm U938, Saint-Antoine Research Centre, Institute of Cardiometabolism and Nutrition, Paris 75012, France. [134]Department of Endocrinology, Diabetology and Reproductive Endocrinology, Assistance Publique-Hôpitaux de Paris, Saint-Antoine University Hospital, National

Reference Center for Rare Diseases of Insulin Secretion and Insulin Sensitivity (PRISIS), Paris, France. [135]Royal Melbourne Hospital Department of Diabetes and Endocrinology, Parkville, Vic, Australia. [136]Walter and Eliza Hall Institute, Parkville, VIC, Australia. [137]University of Melbourne Department of Medicine, Parkville, VIC, Australia. [138]Deakin University, Melbourne, Australia. [139]Department of Epidemiology, Madras Diabetes Research Foundation, Chennai, India. [140]Department of Diabetes and Endocrinology, Guy's and St Thomas' Hospitals NHS Foundation Trust, London, UK. [141]School of Agriculture, Food and Wine, University of Adelaide, Adelaide, Australia. [142]Institut Cochin, Inserm U 10116 Paris, France. [143]Pediatric endocrinology and diabetes, Hopital Necker Enfants Malades, APHP Centre, université de Paris, Paris, France. [144]Department of Medical Genetics, Haukeland University Hospital, Bergen, Norway. [145]Department of Medicine, University of Maryland School of Medicine, Baltimore, MD, USA. [146]Department of Epidemiology, Geisel School of Medicine at Dartmouth, Hanover, NH, USA. [147]Nephrology, Dialysis and Renal Transplant Unit, IRCCS—Azienda Ospedaliero-Universitaria di Bologna, Alma Mater Studiorum University of Bologna, Bologna, Italy. [148]Department of Medical Genetics, AP-HP Pitié-Salpêtrière Hospital, Sorbonne University, Paris, France. [149]Global Center for Asian Women's Health, Yong Loo Lin School of Medicine, National University of Singapore, Singapore, Singapore. [150]Department of Obstetrics and Gynecology, Yong Loo Lin School of Medicine, National University of Singapore, Singapore, Singapore. [151]Kaiser Permanente Northern California Division of Research, Oakland, CA, USA. [152]Department of Epidemiology and Biostatistics, University of California San Francisco, California, CA, USA. [153]National Institute of Diabetes and Digestive and Kidney Diseases, National Institutes of Health, Bethesda, MD, USA. [154]Department of Health Research Methods, Evidence, and Impact, Faculty of Health Sciences, McMaster University, Hamilton, ON, Canada. [155]Ann & Robert H. Lurie Children's Hospital of Chicago, Department of Pediatrics, Northwestern University Feinberg School of Medicine, Chicago, IL, USA. [156]Department of Clinical and Organizational Development, Chicago, IL, USA. [157]American Diabetes Association, Arlington, VA, USA. [158]College of Medicine and Health Sciences, University of Gondar, Gondar, Ethiopia. [159]Global Health Institute, Faculty of Medicine and Health Sciences, University of Antwerp, 2160 Antwerp, Belgium. [160]Department of Medicine and Kovler Diabetes Center, University of Chicago, Chicago, IL, USA. [161]School of Nursing, Faculty of Health Sciences, McMaster University, Hamilton, Canada. [162]Division of Endocrinology, Metabolism, Diabetes, University of Colorado, Boulder, CO, USA. [163]Department of Clinical Medicine, School of Medicine, Trinity College Dublin, Dublin, Ireland. [164]Department of Endocrinology, Wexford General Hospital, Wexford, Ireland. [165]Division of Endocrinology, NorthShore University HealthSystem, Skokie, IL, USA. [166]Department of Medicine, Prtizker School of Medicine, University of Chicago, Chicago, IL, USA. [167]Department of Genetics, Stanford School of Medicine, Stanford University, Stanford, CA, USA. [168]Faculty of Health, Aarhus University, Aarhus, Denmark. [169]Departments of Pediatrics and Medicine and Kovler Diabetes Center, University of Chicago, Chicago, USA. [170]Sanford Research, Sioux Falls, SD, USA. [171]University of Washington School of Medicine, Seattle, WA, USA. [172]Department of Population Medicine, Harvard Medical School, Harvard Pilgrim Health Care Institute, Boston, MA, USA. [173]Department of Medicine, Universite de Sherbrooke, Sherbrooke, QC, Canada. [174]Department of Internal Medicine, Seoul National University College of Medicine, Seoul National University Hospital, Seoul, Republic of Korea. [175]Joslin Diabetes Center, Harvard Medical School, Boston, MA, USA. [176]Charles Bronfman Institute for Personalized Medicine, Icahn School of Medicine at Mount Sinai, New York, NY, USA. [177]Broad Institute, Cambridge, MA, USA. [178]Division of Metabolism, Digestion and Reproduction, Imperial College London, London, UK. [179]Department of Diabetes & Endocrinology, Imperial College Healthcare NHS Trust, London, UK. [180]Department of Diabetology, Madras Diabetes Research Foundation & Dr. Mohan's Diabetes Specialities Centre, Chennai, India. [181]Department of Medicine, Faculty of Medicine and Health Sciences, University of Auckland, Auckland, New Zealand. [182]Auckland Diabetes Centre, Te Whatu Ora Health New Zealand, Auckland, New Zealand. [183]Medical Bariatric Service, Te Whatu Ora Counties, Health New Zealand, Auckland, New Zealand. [184]Oxford NIHR Biomedical Research Centre, University of Oxford, Oxford, UK. [185]University of Cambridge, Metabolic Research Laboratories and MRC Metabolic Diseases Unit, Wellcome-MRC Institute of Metabolic Science, Cambridge, UK. [186]Department of Epidemiology & Public Health, University of Maryland School of Medicine, Baltimore, MD, USA. [187]Department of Internal Medicine, Division of Metabolism, Endocrinology and Diabetes, University of Michigan, Ann Arbor, MI, USA. [188]AdventHealth Translational Research Institute, Orlando, FL, USA. [189]Pennington Biomedical Research Center, Baton Rouge, LA, USA. [190]Yale School of Medicine, New Haven, CT, USA. [191]Faculty of Medicine and Health, University of Sydney, Sydney, NSW, Australia. [192]Department of Endocrinology, Royal Prince Alfred Hospital, Sydney, NSW, Australia. [193]Kaiser Permanente Northwest, Kaiser Permanente Center for Health Research, Portland, OR, USA. [194]Clinial Research, Steno Diabetes Center Copenhagen, Herlev, Denmark. [195]Department of Clinical Medicine, Faculty of Health and Medical Sciences, University of Copenhagen, Copenhagen, Denmark. [196]Department of Endocrinology and Diabetology, University Hospital Düsseldorf, Heinrich Heine University Düsseldorf, Moorenstr. 5, 40225 Düsseldorf, Germany.

