## [Peer Review File · Communications Medicine]

Reviewers' comments:

Reviewer #1 (Remarks to the Author):

In this paper Semple and coworkers aimed at assessing the effects of available/licensed treatments for patients with monogenic severe insulin resistance (SIR) syndromes, including different forms of lipodystrophies and SIR caused by insulin receptor gene mutations. The monogenic forms were stratified according to clinical diagnosis and results of genetic screening. The systematic review started from more than 3000 papers (originals plus large reviews used as source), and forty-two papers were judged good enough for data extraction. Data quality for this review was clearly hampered by the intrinsic nature of SIR, i.e. rarity. This characteristic of SIR makes the attempts to gather enough patients for drug trials very problematic. Therefore the lack of controlled trials is not surprising. Still, Authors were able to collect interesting data for selected drugs, i.e. metreleptin, hIGF1 and thiazolidinediones regarding their impact on dysglycemia, often times present in these patients either as HYPERTENSIVE AND HYPOGLYCEMIA.

General comment. The paper reads well. Introduction, aims, methods and results are clearly presented. Discussion is well focused.

Specific comments. Methods section. In this section (page 5, lines 128-130) the criteria used to exclude papers are clearly described. I was wondering however whether Authors could "relax" the criterion "less than 28 days" for additional search and supplemental results. Because of my own interest in this field I am aware of good papers on short-term therapeutic use of IGF1 in patients with INSR variants. I think that good papers belonging to this subgroup (e.g. Nakae J et al. J Clin Endocrinol Metab 83:542-549, 1998, Longo N et al. J Clin Endocrinol Metab 79:799-805, 1994 or Takahashi Y et al., Diabetologia 40:412-420, 1997 and many others) could reinforce the message and -maybe- allow some considerations on INSR genotype-response to IGF-1. In addition, I think that by extending the search to Dec 31 2022 some new interesting data could be obtained without much effort.

Reviewer #2 (Remarks to the Author):

I was pleased to read the manuscript "systematic review of genotype-stratified treatment for monogenic insulin resistance". I went through the paper's methodology, which impressed me with its accuracy and detail. I confirmed that the review addresses a well-established condition (monogenic insulin resistance), and the question is how to select a precision treatment based on genetic stratification for the prominent phenotype found in these conditions (lipodystrophy). The authors have organized a systematic literature review with a comprehensive approach and precise eligibility and exclusion criteria to answer the question. They have explained all the details in a Protocol that they have registered in PROSPERO and have reported the results according to the PRISMA guidelines.

The literature search is fully documented in a supplement file and has used broad terms with the papers reviewed in parallel by two authors, initially by title and abstract and finally by full-text review of 248 articles. A third author solved the arising conflicts. All the 42 included studies are listed and appraised in terms of internal validity, risk of bias and level of evidence, as documented as a supplement to the paper. The authors also acknowledge unanswered questions regarding

treatments not-reported due to lack of good genotype by treatment evidence, including metformin, SGLT2 inhibitors, GLP1 agonists, rhIGF-1 and bariatric surgery.

In the discussion, authors advance essential explanations about the difficulty of identifying monogenic forms of IR and why expensive treatments like metreleptin have not been trial tested against cheaper alternatives, opening avenues for future research.

I have minor comments or suggestions for amendments that I have listed below:

- Reference 6 needs to be corrected.
- Reference 7 author "Agency, EM" should probably be "EMA."
- Lines 163-165: "Pooled analysis was undertaken for all combinations of genotype and intervention for which sufficient numbers were reported": is there a rule underlying this decision?
- Line 221: "TZDs are potent ligands for the gene product of the PPAR gene" probably better if "TZDs are potent ligands for the product of the PPAR gene"

To conclude, I would recommend the publication of this manuscript.

A homozygous kinase-defective mutation in the insulin receptor gene in a patient with leprechaunism

Y. Takahashi^{1,2}, H. Kadowaki², K. Momomura¹, Y. Fukushima³, T. Orban⁴, T. Okai⁵, Y. Taketani⁵, Y. Akanuma², Y. Yazaki¹, T. Kadowaki¹

¹ Third Department of Internal Medicine, Faculty of Medicine, University of Tokyo, Tokyo, Japan

² Institute for Diabetes Care and Research, Asahi Life Foundation, Tokyo, Japan

³ Saitama Children's Medical Center, Iwatsuki, Japan

⁴ Tawam Hospital, Abu Dhabi, United Arab Emirates

⁵ Department of Obstetrics and Gynecology, Faculty of Medicine, University of Tokyo, Tokyo, Japan

Summary We report a homozygous missense mutation at position 1092 (substitution of glutamine for arginine) in the tyrosine kinase domain of the insulin receptor in a patient with leprechaunism associated with severe insulin resistance and intrauterine growth retardation. Site-directed mutagenesis as well as analyses of the patient's lymphocytes revealed that this mutation causes a marked decrease in tyrosine kinase activity of the insulin receptor without any defect in insulin binding, which causes severe defects in insulin-stimulated glucose transport, glycogen synthesis and DNA synthesis. Thus, this is the first homozygous mutation resulting in a selective-kinase defect of the insulin receptor. Interestingly, the parents who are cousins and are heterozygous for the mutation have type A insulin resistance syndrome. This

correlation between genotype and phenotype in a single pedigree suggests that the severity of the mutation will determine the phenotype. Based upon this assumption, we have been successful in prenatal diagnosis of the fifth child. Furthermore, we have demonstrated the effectiveness of clinical administration of insulin-like growth factor-I (IGF-I) in this patient and in vitro analysis of the patient's skin fibroblasts, suggesting that IGF-I can compensate for insulin action via the IGF-I receptor in a patient almost lacking functional insulin receptors. [Diabetologia (1997) 40: 412–420]

Keywords Insulin resistance syndrome, mutation, genotype, phenotype, tyrosine kinase.

Leprechaunism is the most severe form of the insulin resistance syndrome and is characterized by marked insulin resistance, severe growth retardation, subcutaneous lipodystrophy, acanthosis nigricans, hirsutism and paradoxical fasting hypoglycaemia. Mutations in both alleles of the insulin receptor (IR) gene have been identified in patients with this syndrome, indicating that this is a genetic disease caused by IR gene mutations [1–6]. Type A insulin resistance is

another form of the insulin resistance syndrome, and some patients with the type A syndrome have been found to have mutations in the IR gene [1–4, 7–10]. Patients with the type A syndrome sometimes have mutations in both alleles of the IR gene, while missense mutations in the tyrosine kinase domain of the IR cause type A insulin resistance with only one affected allele [8–16]. Such mutations are called a dominant-negative mutation which decreases tyrosine kinase activity of IR by 75% in vivo. At least three different hypotheses have been proposed to explain the difference in phenotypes of the insulin resistance syndromes associated with IR mutations: 1) severity of insulin resistance; 2) existence of branched pathway; and 3) genetic variation at different loci [1].

The first hypothesis proposes that the severity of the mutation in the IR gene causes the difference in insulin resistance and determines the clinical syndromes; patients with mutations in both alleles of

Received: 25 September 1996 and in revised form: 10 December 1996

Corresponding author: Dr. T. Kadowaki, Third Department of Internal Medicine, Faculty of Medicine, University of Tokyo, 7-3-1 Hongo, Bunkyo-ku, Tokyo, Japan 113

Abbreviations: IR, Insulin receptor; IGF-1, insulin-like growth factor-1; IRI, immunoreactive insulin; PMSF, phenylmethylsulphonyl fluoride; CHO, Chinese hamster ovary cells; IRS-1, insulin-receptor substrate-1.

the IR gene would present a more serious form of the insulin resistance syndrome than those with mutations in only one allele. The second hypothesis proposes that a mutant IR might be impaired in its ability to mediate one biological action while retaining its ability to mediate another; leprechaunism might result from an impairment in both glucose metabolism and the growth-promoting effect mediated by IR, while the type A syndrome might result from a defect only in the metabolic action. The third hypothesis is based upon several publications arguing for the presence of defects in other growth factor receptors in leprechaunism such as the insulin-like growth factor-I (IGF-I) receptor and epidermal growth factor receptor; the type A syndrome might be caused only by mutations in the IR gene, and leprechaunism by mutations in both the IR gene and other growth factor receptor gene(s).

More than 50 kinds of mutations in the IR gene have now been reported [1–22], and understanding of the effects of a homozygous deletion [21], a homozygous nonsense mutation [20, 22] and dominant-negative mutations on the phenotype allows us to predict that the severity of the mutation and the subsequent insulin resistance determine the phenotype. However, there has been no report on an appropriate family with the insulin resistance syndrome showing a more definite correlation between the genotype of IR mutation and the phenotype.

We describe here the genetic analysis of a single pedigree where the two forms of typical insulin resistance syndromes, leprechaunism and type A insulin resistance co-exist. We have also studied a mechanism whereby IGF-I is used as a therapeutic agent for the insulin resistance syndrome at a cellular level. Additionally, we describe the successful prenatal diagnosis of the family's fifth child.

Subjects and methods

The proband was an Arabic boy of 1 year of age, who was born as the fourth child at 36 weeks of gestation with an abnormally low birth-weight (1100 g). He had severe insulin-resistant diabetes (blood glucose 24.8 mmol/l and corresponding immunoreactive insulin (IRI) 55050 pmol/l), severe growth retardation, acanthosis nigricans, subcutaneous lipoatrophy, and other features typical of leprechaunism. His parents were consanguineous and had some typical features of type A insulin resistance syndrome including acanthosis nigricans of the neck and hypertrichosis of the back without obesity or growth retardation. The father, 27 years of age, was diagnosed with overt diabetes mellitus with elevated fasting blood glucose (7.7 mmol/l), moderately increased fasting IRI (438 pmol/l) and haemoglobin A_{1c} (7.3%) but had received no therapy. The mother was 26 years old and had mild fasting hyperinsulinaemia (216 pmol/l), but the corresponding fasting blood glucose (4.0 mmol/l) and haemoglobin A_{1c} (5.1%) were within the

normal range, and she did not have diabetes or polycystic ovary syndrome. The child's eldest brother also had the features of leprechaunism and had died at 7 years of age, while the second and the third boys were clinically normal.

For the genetic analysis of the family, the clinical trial of IGF-I to the proband, and the prenatal diagnosis, informed consent was obtained from both of the parents. The investigations were performed in accordance with the principles of the Declaration of Helsinki.

Study in cultured lymphocytes

We established Epstein-Barr virus-transformed lymphocytes from the proband and his parents. ¹²⁵I-insulin binding study in these cells was performed as previously described by Yamamoto et al. [23], where non-specific binding was assessed in the presence of 1 μmol/l of unlabelled insulin. In vitro autophosphorylation of the insulin receptor partially purified from transformed lymphocytes with wheat-germ agglutinin was performed as previously described [23], in which the content of IRs was normalized by immunoblot analysis before the phosphorylation assay. Tyrosine phosphorylation was detected by immunoblot analysis with anti-phosphotyrosine antibody and ¹²⁵I-protein A.

In vitro tyrosine kinase assay of the IR from cultured lymphocytes was performed with a slight modification of previously described protocols [23–26]. Cells were lysed with ice-cold solubilizing buffer [24] and were subjected to immunoprecipitation with monoclonal anti-IR antibody (α IR-1), anti-mouse IgG and pansorbin or with polyclonal antibody raised against C-terminal peptide of IR (α IRC) [27] and pansorbin at 4°C. Each immune complex was washed three times with ice cold buffer (0.5% Triton X-100, 50 mmol/l HEPES pH 7.4, 10 mmol/l EDTA, 1 mmol/l phenylmethylsulfonyl fluoride (PMSF), 0.3 mol/l NaCl) and resuspended in buffer A (0.1% Triton X-100, 50 mmol/l HEPES pH 7.4, 10 mmol/l MgCl₂, 2 mmol/l MnCl₂, 1 mmol/l sodium orthovanadate, and 1 mmol/l PMSF). Then the amount of IR contained in each immune-complex sample was normalized by ¹²⁵I-insulin binding, and the immune complexes containing the same amount of IR were subjected to in vitro tyrosine-kinase assay as previously described. Samples (50 μl) were incubated with or without 100 nmol/l insulin at room temperature for 30 min, then unlabelled ATP at a final concentration of 50 μmol/l and 1 μCi of γ-³²P ATP was added to each sample with further incubation for 6 min. Histone H2b (10 μg) was added and after 10 min, the reaction was stopped by addition of Laemmli's sample buffer containing β-mercaptoethanol and boiling for 5 min. The samples were then subjected to 15% SDS-PAGE analysis followed by alkaline hydrolysis in the gel as previously described [23–26]. The bands corresponding to histone H2b were excised and counted in a beta-counter.

Direct sequencing analysis

Genomic DNA was extracted from peripheral blood samples of the patient and the parents using the standard method. Each exon of the IR gene was amplified by polymerase chain reaction (PCR) as described by Seino et al. [28, 29]. Direct sequencing analysis was performed as described by Kadowaki et al. [30].

Site-directed mutagenesis

A point mutation was introduced into the expression vector pSV₂-hIR carrying Ullrich's type of normal human insulin receptor cDNA [31] by the method of Kunkel [32] with one base-mismatched oligonucleotide (5'-TCCGTTCTCTGCAG-CAGAGGC-3'). Transfection of the vector into Chinese hamster ovary (CHO) cells (CHO-K1) and selection of clones were performed as described by Kaburagi et al. [24].

Studies on transfected CHO cells (CHO-R1092Q)

Each experiment below was performed independently three times with a representative mutant clone (CHO-R1092Q) and wild-type clone (CHO-hIR), and the insulin binding of each clone expressing mutated or wild-type human IR was almost equal. Also another mutant clone was used to confirm the results of the following experiments.

¹²⁵I-Insulin binding assays were performed as described [24–26]. For the analysis of *in vitro* tyrosine kinase activity, IR purified from CHO cells by wheat-germ agglutinin was immunoprecipitated with α IR-1 antibody, anti-mouse polyclonal antibody and Pansorbin. α IR-1 is specific for human IR and does not precipitate native IR of CHO-K1 cells. The immunoprecipitates, containing approximately the same amount of receptors assessed by insulin binding activity, were prepared for *in vitro* autophosphorylation and tyrosine kinase assay as described above. The equality of content of receptors was confirmed by immunoblotting analysis using α IRC antibody.

Tyrosine phosphorylation of IR and insulin receptor substrate-1 (IRS-1) in intact cells was assessed by immunoprecipitation and immunoblotting with polyclonal anti-phosphotyrosine antibody as previously described [24, 26, 33]. Insulin-dependent glucose uptake was assessed by 2-deoxyglucose uptake assay as described [26, 34]. Measurement of thymidine incorporation into DNA was performed as described by Ando et al. [26]. Glycogen synthase assay was performed as described by Kida and Nyomba [35], and the insulin-dependent glycogen synthase activity was assessed as relative activity (%) against the maximum obtained in the presence of a high concentration of glucose 6-phosphate [35].

Studies in cultured skin fibroblasts

Skin fibroblast cells were obtained from the patient and normal subjects. They were cultured in Dulbecco's modified Eagle's medium (DMEM; Gibco, Grand Island, NY, USA) containing 10% fetal calf serum and antibiotics at 37°C in a 5% CO₂ incubator. Three days after they reached confluence in 10-cm culture dishes, cells were washed and cultured with DMEM containing 0.05% bovine serum albumin (BSA).

Tyrosine phosphorylation of IRS-1 with insulin or IGF-I stimulation was detected by immunoblotting [33]. ¹⁴C-glucose uptake in cultured skin fibroblasts was performed as follows. Cells were cultured in six-well plates as described above and, 3 days after they reached confluence, they were washed and starved with glucose-free DMEM containing 0.05% BSA for 16 hours. Then they were stimulated with various concentrations of insulin or IGF-I for 3 h, followed by the incubation with 0.4 μ Ci of ¹⁴C-glucose and 0.1 mmol/l unlabelled D-glucose for 20 min, then washed quickly with ice cold phosphate buffered saline three times and lysed. The lysates were subjected to liquid scintillation counting.

Clinical administration of IGF-I

Recombinant human IGF-I (rhIGF-I) [36] was supplied from Fujisawa Pharmaceutical Industry, Osaka, Japan. The patient stayed in Japan with his parents during initial treatment for a month, then continued to receive the medication in his own country. IGF-I was initially administered subcutaneously at a dose of 0.1 mg · kg body weight⁻¹ · day⁻¹, and was gradually increased with careful monitoring of blood glucose. It should be noted that the patient is only a small child being fed milk, therefore we did not measure his daily caloric intake precisely.

After 6 month's treatment he received a variety of medical investigations in his own country including physical examination, chest X-ray, echograms of heart and abdomen, and blood examination.

Prenatal diagnosis in the fifth child. During pregnancy with the fifth child, the parents hoped to have genetic information before birth. We obtained informed consent from the parents and performed amniocentesis, sampling the fluid at 17 weeks of gestation and culturing it for 7 days. Following extraction of genomic DNA and PCR amplification of the region spanning exon 18, Pst I digestion was performed on the basis of the genetic study of the family.

Results

Studies in cultured lymphocytes. The ¹²⁵I-insulin binding results are shown in Figure 1 A. Binding was 23.0% in the proband's cells, 14.0% in the father's and 13.6% in the mother's, which were within the normal range (mean \pm SD, 26.6% \pm 7.1%, *n* = 7). There was no significant change in the affinity of the IR in the proband's and parent's cells. As shown in Figure 1 B, autophosphorylation of IR was severely decreased in the proband's cells which were barely detected by immunoblotting. In the father's cells and the mother's cells, there was a slight difference; autophosphorylation of IR was decreased by 70% in the father's cells, and by 40% in the mother's cells. Tyrosine kinase activity of IR in α IRC immune-complex as shown in Figure 1 C was markedly decreased in the proband's cells (\geq 90%), and mildly or moderately decreased in the father's (70–80%) and mother's (50–60%). There was no significant difference in the results found using a different antibody α IR-1 (data not shown). Immunoblot analysis confirmed that each sample contained approximately the same amount of IR in each experiment (data not shown).

Detection of mutation in the IR gene. Direct sequencing analysis revealed that the proband was homozygous for a novel missense mutation in the tyrosine kinase domain (substitution of glutamine (CAG) for arginine (CGG) at position 1092, which amino acid number corresponds to that of Ebina's type IR cDNA [37]), and the parents were heterozygous for the mutation (data not shown). Taking advantage of the fact that this mutation creates another recognition site for restriction endonuclease Pst I

Fig. 1A–C. Studies in Epstein-Barr virus-transformed lymphocytes. **A** ¹²⁵I-insulin binding study with transformed lymphocytes from the proband, the parents and a normal subject. The experiment was repeated independently three times and a representative result is shown. **B** In vitro autophosphorylation of lectin-purified insulin receptor. Tyrosine phosphorylation of the β subunit was detected by anti-phosphotyrosine antibody and ¹²⁵I-protein A. **C** In vitro tyrosine kinase activity of insulin receptors immunopurified from the transformed lymphocytes. The tyrosine kinase activity was measured as ³²P incorporation into histone H2b. The amount of IR in each sample was normalized as described in methods. C: normal control subjects; PT: patient; F: father; M: mother

(CTGCAG), the PCR products spanning exon 18 were digested with Pst I to confirm the result of sequencing analysis (data not shown).

Studies in CHO cells. ¹²⁵I-insulin binding study revealed that the expression of the mutant IR in CHO-R1092Q cells (specific insulin binding: 55%) was almost equal to that of the wild-type IR expressing in CHO-hIR cells (specific insulin binding: 60%), and the following Scatchard analysis indicated that the mutated IR has no difference in insulin binding affinity as compared to the normal IR (data not shown). Consistent with this, the same amount of protein fraction purified by wheat-germ agglutinin from these cells contained an equal amount of IR (Fig. 2A). Insulin-dependent tyrosine-kinase activity of the mutant IR (Fig. 2C) as well as autophosphorylation (Fig. 2B) was severely decreased by approximately 90%. Thus, severely impaired tyrosine kinase activity of the mutant IR with no significant effect on insulin binding was apparent and consistent with the study on cultured lymphocytes of the proband. Tyrosine phosphorylation of IRS-1 in CHO-R1092Q cells was severely impaired, and almost equal to that in untransfected CHO-K1 cells (Fig. 2D).

Insulin-stimulated 2-deoxyglucose uptake, thymidine incorporation into DNA, and glycogen synthase activation were all markedly impaired in

CHO-R1092Q cells as compared to CHO-hIR cells (Fig. 3A–C, respectively). The insulin-stimulated maximal response of 2-deoxyglucose uptake and glycogen synthase activation were severely decreased in CHO-R1092Q cells. In thymidine incorporation, the maximal response was approximately the same, but the dose of insulin required to reach the half maximal response in CHO-R1092Q cells was about 100 times higher than that in CHO-hIR cells. These were consistent with the decreased tyrosine kinase activity of the mutant IR and the severely impaired tyrosine phosphorylation of IRS-1 (Fig. 2). These results suggest that the homozygous mutation in the tyrosine kinase domain of the IR is responsible for the defect in glucose metabolism and in the growth retardation of the proband.

Studies in cultured skin fibroblasts. Immunoblot analysis revealed that the amount of IR was not significantly different among the cells of the normal control and the proband (data not shown). Insulin-stimulated phosphorylation of IRS-1 was markedly decreased in the proband, whereas IRS-1 was almost normally phosphorylated by IGF-I stimulation (data not shown). To confirm that the actions of insulin and IGF-I in patient's cells correlated with the IRS-1 phosphorylation, we performed a ¹⁴C-glucose uptake assay to assess insulin and IGF-I action (Fig. 4). In the control cells, the maximal response induced by insulin was obtained at 10 nmol/l insulin, which was approximately the same magnitude as the maximum induced by IGF-I. In contrast, ¹⁴C-glucose uptake stimulated by insulin in the patient's cells was markedly impaired, although sufficient glucose uptake was obtained with the stimulation of IGF-I.

Clinical course of IGF-I treatment. In the initial 1-month treatment in Japan, the low dose of rhIGF-I (0.1 mg · kg⁻¹ · day⁻¹) was effective in lowering the blood glucose level at first, but 2 or 3 days after the dose-up, more rhIGF-I was required for blood glucose control due to the increased food intake. For that reason, rhIGF-I was administered finally at a

Fig 2A–D. Tyrosine kinase activity of the mutant receptor expressed on CHO cells. **A** Immunoblot analysis for quantification of lectin-purified human type IR immunoprecipitated by α IR-1 antibody, which was subjected to experiments B and C. Lane 1: CHO-K1 (untransfected) Lane 2: CHO-hIR (wild-type) Lane 3: CHO-R1092Q (mutant-type). **B** In vitro autophosphorylation of the IR β subunit. **C** In vitro tyrosine kinase activity of the IR. **D** Phosphorylation of IRS-1 in intact CHO cells. Phosphorylation of IRS-1 was markedly reduced in CHO-R1092Q, almost as little as that in CHO-K1 cells. Autophosphorylation of the receptor β subunit in CHO-R1092Q was also impaired, but was slightly larger than that of CHO-K1, which was consistent with the data in panel B

genotype and phenotype in this family, we predicted that the fifth child would not have leprechaunism. In fact, the newborn was a heterozygote and did not show leprechaunism.

Discussion

To our knowledge, our case is the first to show the co-existence of two types of insulin resistance syndrome in a single pedigree. The homozygous patient with leprechaunism, and the heterozygous parents with type A insulin resistance; tyrosine kinase activity of IR in the proband's cells was markedly decreased, and that in the parent's cells was mildly or moderately decreased. Thus, a tight correlation among genotype, phenotype and tyrosine kinase activity of IR was found in this family. It is of note that this family is consanguineous, and the members have a relatively common genetic background. Therefore, our observations are highly suggestive that the severity of mutation will determine the phenotype of insulin resistance syndrome associated with IR mutations. In fact, we correctly predicted the phenotype of the newborn in prenatal diagnosis. The residue 1092 on a connecting loop between the two lobes of the β subunit, which interact to make an open or closed form, has been suggested to be implicated in peptide substrate specificity [39]. Based upon the structure and the fact that tyrosine kinase activity was decreased by 50–75% in the heterozygous parents, the mutation is likely to act in a dominant-negative fashion. It will be of interest to see whether the newborn will develop type A insulin resistance in the future.

dose of $1 \text{ mg} \cdot \text{kg}^{-1} \cdot \text{day}^{-1}$, which was the maximum of our protocol. As indicated in Table 1, IGF-I administration was remarkably effective; the patient's fasting blood glucose levels, haemoglobin A_{1c} values and serum immunoreactive insulin levels were markedly improved. Moreover, his body weight increased, and his development was promoted. No adverse effects including allergy, oedema, hypoglycaemic episodes, and organomegaly were seen during the 6 months. Acanthosis nigricans and other physical features of leprechaunism did not change, nor did diabetic complications develop during this period.

Prenatal diagnosis. To determine the genotype of the fetus, Pst I digestion for PCR products was performed instead of direct sequencing [38], and the fetus was diagnosed as a heterozygote for the mutation (not shown). Based upon the correlation between

Fig. 3A-C. Biological action of insulin in intact CHO cells. **A** Two-deoxyglucose uptake. **B** Thymidine incorporation into DNA. **C** Glycogen synthase assay. All data are representative of the experiments performed independently at least three times, and the results are assessed by fold stimulation

Many patients with leprechaunism have mutations in the α subunit of the IR which decrease the binding affinity to insulin and/or impair IR processing [1-4]. In such cases, the intensity of signals from mutated IR to intracellular messenger molecules are

diminished due to the defect in ligand-dependent activation of tyrosine kinase in the intact β subunit and/or a decreased amount of IR on the cell surface. Three cases with leprechaunism lacking in IR have recently been reported [20-22], and insulin signalling mediated by the IR will be totally knocked out in those cases. In our case, insulin binding affinity and the amount of IR on the cell surface was within the normal range, and the results in the study of site-directed mutagenesis suggest that a selective defect in tyrosine kinase causing a severe defect in both metabolic action and growth-promoting action will be sufficient to cause leprechaunism. The case of a compound heterozygote with type A insulin resistance has been reported with a nonsense mutation in one allele and a missense mutation in the other, both of which lie in the tyrosine kinase domain [19]. In the report, tyrosine kinase activity of IR prepared from the patient's cells was decreased by 75% as compared to that from the same number of normal control cells. So we speculate that the degree of defect in insulin signalling in that case was not significantly different from those in our heterozygous case or other reported simple heterozygotes with kinase-defective mutation.

Fig. 4A-C. ^{14}C -glucose uptake analysis in cultured skin fibroblasts. **A** shows the basal and maximal uptake of ^{14}C -glucose in cells of the control subject (C, ●) and the patient (Pt, ○). **B** and **C** show % response at indicated doses of insulin (**B**) or IGF-I (**C**) in cells of the control subject and the patient. In the control cells (●), the maximal uptake with the stimulation of insulin was almost the same as the maximum with the stimulation of IGF-I, while the marked impairment in insulin-stimulated ^{14}C -glucose uptake was clear in the patient's cells (○). Slight increase of ^{14}C -glucose uptake by insulin in the patient's cells will be via the IGF-I receptor bound to insulin. Data are representative of three experiments

Table 1. Summary of IGF-I treatment for 6 months. The clinical data were summarized on the patient's growth, development and diabetes

	IGF-I Before	1 mg · kg ⁻¹ · day ⁻¹	
		After 1 month	After 6 months
Age	1y10M	1y11m	2y4m
Height (cm)	63		70
Weight (kg)	4.8	5.3	6.3
Fasting blood glucose (mmol/l)	17.2	8.8	7.7
FIRI (pmol/l)	11880	2532	588
HbA _{1c} (%)	10	9.3	8
Development	Cannot sit up or speak	Sits up with a little help	Walks with a little help and speaks

FIRI, Fasting immunoreactive insulin

We have shown the slight difference in autophosphorylation and tyrosine kinase activity of IR between the parents (Fig. 1). No additional mutation has been found in the father. RT-PCR and allele-specific oligonucleotide hybridization analysis using lymphocyte mRNAs revealed an almost equal expression of the wild-type and mutant-type alleles in both of the parents (data not shown). In addition, all the receptors expressed in lymphocytes were the type described by Ullrich et al. [31]. Thus, the difference cannot be explained by the mRNA level or alternative splicing in the IR gene. Some diversity in receptor processing may modify the kinase activity, but we do not yet know the mechanism of the difference. Importantly, insulin-stimulated tyrosine phosphorylation of IRS-1 was moderately decreased but was not apparently different between the parents' skin fibroblasts (data not shown). Therefore, the difference in tyrosine kinase activity does not explain the development of diabetes only in the father despite the fact that the parents had the same heterozygous mutation. Although the dominant-negative mutation in IR is thought to be linked to type A syndrome, the development of diabetes may also be affected by additional genetic and environmental factors such as insulin secretory capacity and lifestyle apart from the IR mutation.

How can such patients almost lacking functional IR be born? We cannot completely exclude the possibility that some insulin action(s) may be mediated without the kinase activity, although this is less likely considering that the absolute requirement of IR tyrosine kinase has been well-documented for most of the insulin actions. One plausible explanation is the residual tyrosine kinase of the mutant receptor shown in Figures 1 and 2, and the patient's IR function in vivo may not be totally abolished. Another likely explanation is the compensation by the IGF-I receptor. IGF-I is known to have some insulin action in intact

cells, and its receptor can phosphorylate IRS-1, which is thought to play a key role in insulin signalling [40, 41]. In fact, Shc protein has been reported to mediate insulin action and to be involved in IGF-I receptor signalling [42]. IRS-1 and Shc bind to the GRB2/ASH adapter protein when activated by insulin receptor or IGF-I receptor tyrosine kinase, thus the two molecules are involved in a common pathway to the activation of ras mediated by the insulin receptor and the IGF-I receptor. The IGF-I receptor is expressed early in the fetal period, and can bind to insulin although its affinity is far lower than that of the IR. After starting insulin secretion, it is possible that serum insulin level will reach 10 to 100 times that of normal and fetal insulin will bind to the IGF-I receptor. In fact, clinical administration of IGF-I to the proband was remarkably effective in glucose metabolism, growth and development. ¹⁴C-glucose uptake assay in skin fibroblasts as well as IGF-I-induced tyrosine phosphorylation of IRS-1 revealed a compensatory effect of IGF-I in glucose metabolism at the cellular level of the patient, which is contrary to a recent report arguing for a postreceptor defect of IGF-I signalling in leprechaunism [43]. Therefore, we concluded that a common pathway via the IGF-I receptor was intact in this patient, which made a substantial contribution to his survival. It remains to be further investigated whether the heterodimerization of the mutant receptor with IGF-I receptor works in vivo.

Two patients with a homozygous nonsense mutation [20, 22] and another patient with a homozygous deletion [21] in the IR gene did not die in the fetal period and both showed leprechaunism. Also, targeted ablation of the IR gene in mice resulted in early neonatal death [44]. These appear consistent with our hypothesis on the fetal life of leprechaunism. Notably, the compensatory mechanism in disruption of the IR is apparently insufficient as compared to the IRS-1 knockout mice which are rescued by IRS-2 [45, 46], indicating more redundancy in intracellular signalling molecules and more critical roles for the cell surface IR in life.

Finally, we have shown the usefulness of rhIGF-I, but the dose required for the patient's treatment was quite high. We do not precisely know the long-term effect of IGF-I [36, 47] and we should carefully continue the therapy for the patient with life-threatening severe insulin resistance.

Acknowledgements. We are indebted to Dr. S.I. Taylor for helpful comments, and to Dr. Y. Gotoda and his medical staff for intensive medical care during initial IGF-I treatment for the proband in Japan.

Supported in part by a grant (192125) from the Juvenile Diabetes Foundation International to T.K.

References

1. Taylor SI (1992) Molecular mechanism of insulin resistance. Lessons from patients with mutations in the insulin receptor gene. *Diabetes* 41: 1473–1490
2. Flier JS (1992) Syndromes of insulin resistance. From patient to gene and back again. *Diabetes* 41: 1207–1219
3. Moller DE, Flier JS (1991) Mechanism of disease: insulin resistance mechanisms, syndromes, and implications. *N Engl J Med* 325: 938–948
4. O’Rahilly S, Moller DE (1992) Mutant insulin receptors in syndromes of insulin resistance. *Clin Endocrinol* 36: 121–132
5. Kadowaki T, Bevins CL, Cama A et al. (1988) Two mutant alleles of the insulin receptor gene in a patient with extreme insulin resistance syndrome. *Science* 240: 787–790
6. Yoshimasa Y, Seino S, Whittaker J et al. (1988) Insulin-resistant diabetes due to a point mutation that prevents insulin receptor processing. *Science* 240: 784–787
7. Kahn CR, Flier JS, Bar RS et al. (1976) The syndromes of insulin resistance and acanthosis nigricans: insulin receptor disorder in man. *N Engl J Med* 294: 739–745
8. Moller DE, Cohen O, Yamaguchi Y et al. (1994) Prevalence of mutations in the insulin receptor gene in subjects with features of the type A syndrome of insulin resistance. *Diabetes* 43: 247–255
9. Moller DE, Flier JS (1988) Detection of an alteration in the insulin-receptor gene in a patient with insulin resistance, acanthosis nigricans, and the polycystic ovary syndrome (type A insulin resistance). *N Engl J Med* 319: 1526–1529
10. Odawara M, Kadowaki T, Yamamoto R et al. (1989) Human diabetes associated with a mutation in the tyrosine kinase domain of the insulin receptor. *Science* 245: 66–68
11. Kadowaki T, Kadowaki H, Rechler MM et al. (1990) Five mutant alleles of the insulin receptor gene in patients with genetic forms of insulin resistance. *J Clin Invest* 86: 254–264
12. Cama A, Sierra ML, Ottini L et al. (1991) A mutation in the tyrosine kinase domain of the insulin receptor associated with insulin resistance in an obese woman. *J Clin Endocrinol Metab* 73: 894–901
13. Kim H, Kadowaki H, Sakura H et al. (1992) Detection of mutations in the insulin receptor gene in patients with insulin resistance by analysis of single-stranded conformational polymorphisms. *Diabetologia* 35: 261–266
14. Taira M, Hashimoto N, Shimada F et al. (1989) Human diabetes associated with a deletion of the tyrosine kinase domain of the insulin receptor. *Science* 245: 63–66
15. Haruta T, Takata Y, Iwanishi M et al. (1993) Ala1048 → Asp mutation in the kinase domain of insulin receptor causes defective kinase activity and insulin resistance. *Diabetes* 42: 1837–1844
16. Iwanishi M, Haruta T, Takata Y et al. (1993) A mutation (Trp 1193 → Leu 1193) in the tyrosine kinase domain of the insulin receptor associated with type A syndrome of insulin resistance. *Diabetologia* 36: 414–422
17. Accili D, Frapier C, Mosthaf L et al. (1989) A mutation in the insulin receptor gene that impairs transport of the receptor to the plasma membrane and causes insulin-resistant diabetes. *EMBO J* 8: 2509–2517
18. Shimada F, Taira M, Suzuki Y et al. (1990) Insulin-resistant diabetes associated with partial deletion of insulin-receptor gene. *Lancet* 335: 1179–1181
19. Kusari J, Takata Y, Hatada E, Freidenberg G, Kolterman O, Olefsky JM (1991) Insulin resistance and diabetes due to different mutation in the tyrosine kinase domain of both insulin receptor gene alleles. *J Biol Chem* 266: 5260–5267
20. Krook A, Brueton L, O’Rahilly S (1993) Homozygous nonsense mutation in the insulin receptor gene in infant with leprechaunism. *Lancet* 342: 277–278
21. Wertheimer E, Lu S, Backeljauw PF, Davenport ML, Taylor SI (1993) Homozygous deletion of the human insulin receptor gene results in leprechaunism. *Nature Genetics* 5: 71–73
22. Psiachou H, Mitton S, Alagband ZJ, Hone J, Taylor SI, Sinclair L (1993) Leprechaunism and homozygous nonsense mutation in the insulin receptor gene. *Lancet* 342: 924
23. Yamamoto R, Shibata T, Tobe K et al. (1990) Defect in tyrosine kinase activity of the insulin receptor from a patient with insulin resistance and acanthosis nigricans. *J Clin Endocrinol Metab* 70: 869–878
24. Kaburagi Y, Momomura K, Yamamoto-Honda R et al. (1993) Site-directed mutagenesis of the juxtamembrane domain of the human insulin receptor. *J Biol Chem* 268: 16610–16622
25. Yamamoto-Honda R, Koshio O, Tobe K et al. (1990) Phosphorylation state and biological function of a mutant human insulin receptor Val 996. *J Biol Chem* 265: 14777–14783
26. Ando A, Momomura K, Tobe K et al. (1992) Enhanced insulin-induced mitogenesis and mitogen-activated protein kinase activities in mutant insulin receptors with substitution of two COOH-terminal tyrosine autophosphorylation sites by phenylalanine. *J Biol Chem* 267: 12788–12796
27. Izumi T, Saeki Y, Akanuma Y, Takaku F, Kasuga M (1993) Requirement for receptor-intrinsic tyrosine kinase activities during ligand-induced membrane ruffling of KB cells. *J Biol Chem* 263: 10386–10393
28. Seino S, Seino M, Bell GI (1990) Human insulin receptor gene: partial sequence and amplification of exons by polymerase chain reaction. *Diabetes* 39: 123–128
29. Seino S, Seino M, Nishi S, Bell GI (1989) Structures of the human insulin receptor gene and its characterization of its promoter. *Proc Natl Acad Sci USA* 86: 114–118
30. Kadowaki T, Kadowaki H, Taylor SI (1990) A nonsense mutation causing decreased levels of insulin receptor mRNA: detection by a simplified technique for direct sequencing of genomic DNA amplified by polymerase chain reaction. *Proc Natl Acad Sci USA* 87: 658–662
31. Ullrich A, Bell JR, Chen EY et al. (1985) Human insulin receptor and its relationship to the tyrosine kinase family of oncogenes. *Nature* 313: 756–761
32. Kunkel TA (1985) Rapid and efficient site-specific mutagenesis without phenotypic selection. *Proc Natl Acad Sci USA* 82: 488–492
33. Momomura K, Tobe K, Seyama Y, Takaku F, Kasuga M (1988) Insulin-induced tyrosine-phosphorylation in intact rat adipocytes. *Biochem Biophys Res Commun* 155: 1181–1186
34. Chou CK, Dull TJ, Russel DS et al. (1987) Human insulin receptors mutated at the ATP-binding site lack protein tyrosine kinase activity and fail to mediate postreceptor effects of insulin. *J Biol Chem* 262: 1842–1847
35. Kida Y, Nyomba BL (1991) Defective insulin response of cyclic adenosine monophosphate-dependent protein kinase in insulin-resistant humans. *J Clin Invest* 87: 673–679
36. Kuzuya H, Matsuura N, Sakamoto M et al. (1993) Trial of insulinlike growth factor I therapy for patients with extreme insulin resistance syndromes. *Diabetes* 42: 696–705
37. Ebina Y, Ellis L, Jarnagin K et al. (1985) The human insulin receptor cDNA. The structural basis for hormone-activated transmembrane signalling. *Cell* 40: 747–758

38. Longo N, Langley SD, Griffin LD, Elsas LJ (1995) Two mutations in the insulin receptor gene of a patient with leprechaunism: application to prenatal diagnosis. *J Clin Endocrinol Metab* 80: 1496–1501
39. Hubbard SR, Wei L, Ellis L, Hendrickson WA (1994) Crystal structure of the tyrosine kinase domain of the human insulin receptor. *Nature* 372: 746–754
40. Sun XJ, Rothenberg P, Kahn CR et al. (1991) Structure of the insulin receptor substrate IRS-1 defines a unique signal transduction protein. *Nature* 352: 73–77
41. Kadowaki T, Koyasu S, Nishida E (1987) Tyrosine phosphorylation of common and specific sets of cellular proteins rapidly induced by insulin. *J Biol Chem* 262: 1282–1287
42. Sasaoka T, Rose DW, Jhun BH, Saltiel AR, Draznin B, Olefsky JM (1994) Evidence for a functional role of Shc proteins in mitogenic signalling induced by insulin, insulin-like growth factor-1, and epidermal growth factor. *J Biol Chem* 269: 13689–13694
43. Backeljauw PF, Alves C, Eidson M, Cleveland W, Underwood LE, Davenport ML (1994) Effect of intravenous insulin-like growth factor I in two patients with leprechaunism. *Pediatr Res* 36: 749–754
44. Accili D, Drago J, Lee EJ et al. (1996) Early neonatal death in mice homozygous for a null allele of the insulin receptor gene. *Nat Genet* 12: 106–109
45. Tamemoto H, Kadowaki T, Tobe K et al. (1994) Insulin resistance and growth retardation in mice lacking insulin receptor substrate-1. *Nature* 372: 182–186
46. Araki E, Lipes MA, Patti M-E et al. (1994) Alternative pathway of insulin signalling in mice with targeted disruption of the IRS-1 gene. *Nature* 372: 186–190
47. Kolaczynski JW, Caro JF (1994) Insulin-like growth factor-1 therapy in diabetes: physiologic basis, clinical benefits, and risks. *Ann Intern Med* 120: 47–55

Note added in proof: The patient is now 5 years old and still on IGF-I treatment. His weight is 10.5 kg, and his height 82 cm. Physical examination apart from the features of leprechaunism is unremarkable.

Response to reviewers' comments

Reviewer #1

In this paper Semple and coworkers aimed at assessing the effects of available/licensed treatments for patients with monogenic severe insulin resistance (SIR) syndromes, including different forms of lipodystrophies and SIR caused by insulin receptor gene mutations. The monogenic forms were stratified according to clinical diagnosis and results of genetic screening. The systematic review started from more than 3000 paper (originals plus large reviews used as source), and forty-two papers were judged good enough for data extraction. Data quality for this review was clearly hampered by the intrinsic nature of SIRs., i.e. rarity. This characteristic of SIR makes the attempts to gather enough patients for drug trials very problematic. Therefore the lack of controlled trials is not surprising. Still, Authors were able to collect interesting data for selected drugs, i.e. metreleptin, rhIGF1 and thiazolidinediones regarding their impact on dysglycemia, often times present in these patients either as HYPER- AND HYPOglycemia.

General comment. The paper reads well. Introduction, aims, methods and results are clearly presented. Discussion is well focused.

Specific comments. Methods section. In this section (page 5, lines 128-130) the criteria used to exclude papers are clearly described. I was wondering however whether Authors could "relax" the criterium "less than 28 days" for additional search and supplemental results. Because of my own interest in this field I am aware of good papers on short-term therapeutic use of IGF1 in patients with INSR variants. I think that good papers belonging to this subgroup (e.g. Nakae J et al. J Clin Endocrinol Metab 83:542-549, 1998, Longo N et al. J Clin Endocrinol Metab 79:799-805, 1994 or Takahashi Y et al., Diabetologia 40:412-420, 1997 and many others) could reinforce the message and -maybe- allow some considerations on INSR genotype-response to IGF-1. In addition, I think that by extending the search to Dec 31 2022 some new interesting data could be obtained without much effort.

Author's Response: This systematic review was one component of a large initiative to review precision medicine in various forms of diabetes. Our search strategy was reached through consensus during the first semester of 2021 with the wider group, and in order to maintain consistency across the multiple systematic reviews, we do not feel that it is appropriate to update the search through the end of 2022. The likelihood of identifying new papers that would change the conclusion of the current report is minimal, and such a massive search update would imply many weeks of work.

Regarding the duration of treatment, we identified 5 articles that were excluded due to a treatment duration <28 days:

10.1186/s13098-019-0477-z	Lewandowski 2019	Metformin paradoxically worsens insulin resistance in SHORT syndrome
10.2337/dc16-0635	Robinson 2016	Management of diabetic ketoacidosis in severe insulin resistance
10.1203/00006450-199412000-00012	Backeljauw 1994	Effect of intravenous insulin-like growth factor I in two patients with leprechaunism
10.1177/0897190016645036	Moore 2017	Treatment of Diabetic Ketoacidosis With Intravenous U-500 Insulin in a Patient With Rabson-Mendenhall Syndrome: A Case Report
10.1530/eje.0.1310251	Zenobi 1994	Beneficial metabolic effects of insulin-like growth factor I in patients with severe insulin-resistant diabetes type A

COMMSMED-23-0277: Systematic review of genotype-stratified treatment for monogenic insulin resistance

We chose to exclude papers reporting data after <28 days of intervention because of our focus on clinically important outcomes such as HbA1c. We acknowledge that reports of shorter duration treatments, especially for IGF-1, have provided proof of concept for acute beneficial metabolic actions for interventions, but these have already been well reviewed for rhIGF1 (e.g. *Int. J. Mol. Sci.* 2018, 19(5), 1268; <https://doi.org/10.3390/ijms19051268>; PMID: [29695048](https://pubmed.ncbi.nlm.nih.gov/29695048/)). We have commented on this in the revised manuscript as copied below:

“While our review focused only on studies reporting treatment durations of greater than 28 days so that changes in HbA1c could be assessed, many shorter duration studies of IGF-1 therapy in people with INSR mutations exist, and have yielded important mechanistic insights and provide proof on concept for acute metabolic benefits (summarized in [1]).” [Lines 311-315]

Prompted by the reviewer’s helpful observations and citations we did, however, note that some papers on INSR and IGF-1 were missed by the initial search strategy. We therefore conducted a supplementary manual review of the bibliography of the above key review (PMID: [29695048](https://pubmed.ncbi.nlm.nih.gov/29695048/)) to identify additional papers meeting our inclusion criteria. This identified 7 papers not previously identified by our search strategy, although only 2 of these met inclusion criteria for the systematic review. Data from these papers, including 1 patient each) are included in the updated analyses. Of note, of the 3 papers specifically mentioned by the reviewer, only 1 met criteria for inclusion (Nakae 1998 was excluded because glycemia was not reported prior to initiation of IGF-1; Longo 1994 was excluded because molecular genetics were not reported; Takahashi 1997 is now included).

For the reviewer’s interest, here are all rhIGF1 papers considered based on additional citation review:

Author	Treatment Duration of rhIGF-I	SIR Syndrome (n)	Included / Excluded (new or in original manuscript)	Exclusion reason
Quin et al., 1990 [23]	1 dose	RMS (1)	Excluded (new)	<28 days
Schoenle et al., 1991 [24]	2 doses	Type A IR (3)	Excluded (new)	<28 days
Hussain et al., 1993 [22]	4 days, BID	Type A IR (1)	Excluded (new)	<28 days
Kuzuya et al., 1993 [73]	Up to 16 months	Type A IR (6) DS (2) Lipodystrophy (2) Other (1)	Included (original)	
Morrow et al., 1994 [27]	3–4 weeks	Type A without INSR mutations (2)	Excluded (new)	Wrong patient population (not INSR)
Backeljauw, 1994 [75]	66 h and 62 h	DS (2)	Excluded (original)	<28 days

COMMSMED-23-0277: Systematic review of genotype-stratified treatment for monogenic insulin resistance

Author	Treatment Duration of rhIGF-I	SIR Syndrome (n)	Included / Excluded (new or in original manuscript)	Exclusion reason
Zenobi, 1994 [25]	5 days	Type A IR (2)	Excluded (original)	<28 days
Longo et al., 1994 [63]	16 months	RMS (1)	Excluded (new)	No molecular genetics
Nakashima et al., 1995 [28]	9 months	Type A IR (1)	Included (new)	
Vestergaard et al., 1997 [26]	2 weeks high dose 10 weeks low dose	SIR (4)	Included (original)	
Takahashi et al., 1997 [32]	6 months	Leprechaunism (1)	Included (new)	
Nakae et al., 1998 [29]	6 years and 10 months	DS (?) or RMS (?) (1 patient, different times)	Excluded (new)	No outcome reported (no A1c pre-treatment)
Kitamei et al., 2004 [78]				No outcome reported (no A1c pre-treatment) – same patient as Nakae et al
Jo et al., 2013 [77]	Withdrawal of rhIGF-I at 18 years, due to DKA and start of high dose insulin		Included (original)	
Regan et al., 2010 [79]	16 weeks	Type A IR (5)	Included (original)	
Weber et al., 2014 [30]	16 months (to death at 35 months)	DS (1)	Included (original)	
De Kerdanet et al., 2015 [48]	8.7 years; 2 years	DS (1)	Included (original)	
Carmody et al., 2016 [31]	5 years	RMS (1)	Included (original)	

COMMSMED-23-0277: Systematic review of genotype-stratified treatment for monogenic insulin resistance

Reviewer #2

I was pleased to read the manuscript "systematic review of genotype-stratified treatment for monogenic insulin resistance". I went through the paper's methodology, which impressed me with its accuracy and detail. I confirmed that the review addresses a well-established condition (monogenic insulin resistance), and the question is how to select a precision treatment based on genetic stratification for the prominent phenotype found in these conditions (lipodystrophy). The authors have organised a systematic literature review with a comprehensive approach and precise eligibility and exclusion criteria to answer the question. They have explained all the details in a Protocol that they have registered in PROSPERO and have reported the results according to the PRISMA guidelines.

The literature search is fully documented in a supplement file and has used broad terms with the papers reviewed in parallel by two authors, initially by title and abstract and finally by full-text review of 248 articles. A third author solved the arising conflicts. All the 42 included studies are listed and appraised in terms of internal validity, risk of bias and level of evidence, as documented as a supplement to the paper. The authors also acknowledge unanswered questions regarding treatments not-reported due to lack of good genotype by treatment evidence, including metformin, SGLT2 inhibitors, GLP1 agonists, rhIGF-1 and bariatric surgery.

In the discussion, authors advance essential explanations about the difficulty of identifying monogenic forms of IR and why expensive treatments like metreleptin have not been trial tested against cheaper alternatives, opening avenues for future research.

I have minor comments or suggestions for amendments that I have listed below:

- Reference 6 needs to be corrected.*
- Reference 7 author "Agency, EM" should probably be "EMA."*

Author's Response: These references were removed as they were not needed, on review.

- Lines 163-165: "Pooled analysis was undertaken for all combinations of genotype and intervention for which sufficient numbers were reported": is there a rule underlying this decision?

Author's Response: In short, no, this was not based on any formal consideration of power. The reviewer will appreciate that a large number of gene x treatment combinations featured 0,1 or 2 people only (Supplemental table 3), and we did not judge statistical analysis meaningful for these in general. For the combinations where we did have larger numbers we undertook the pooled analysis shown in supplementary table 4. In this table we did also include two treatments in lipodystrophy with only 2 subjects to complement the other more complete dataset for lipodystrophy treatments. ALL gene x treatment combinations are illustrated for individual subjects in supplementary figures.

- Line 221: "TZDs are potent ligands for the gene product of the PPARG gene" probably better if "TZDs are potent ligands for the product of the PPARG gene"

Author's Response: This was an error – now edited as suggested.

REVIEWERS' COMMENTS:

Reviewer #1 (Remarks to the Author):

I am satisfied by Authors reply to my comments.

Fabrizio Barbetti

Reviewer #2 (Remarks to the Author):

I thank the authors for addressing the reviewers' comments on their manuscript "Systematic review of genotype-stratified treatment for monogenic insulin resistance". I am personally satisfied with the answers and modifications of the manuscript, addressing both my comments and the other reviewer's comments.

I believe the manuscript brings practical information for the care of monogenic RIS patients by appraising the sparse existing literature and synthesising existing evidence. I would be rather pleased to see the manuscript in print.